# Single-cell transcriptomics reveal extracellular vesicles secretion with a cardiomyocyte proteostasis signature during pathological remodeling

Eric Schoger [1,2,3,13], Federico Bleckwedel[1,2,13], Giulia Germena [2,4], Cheila Rocha [4], Petra Tucholla[1,2], Izzatullo Sobitov[1,2], Wiebke Möbius [5], Maren Sitte [6], Christof Lenz [7,8], Mostafa Samak[2,4], Rabea Hinkel[2,4,9], Zoltán V. Varga[10,11], Zoltán Giricz [10,11], Gabriela Salinas[6], Julia C. Gross [12] & Laura C. Zelarayán [1,2,3✉]

Aberrant Wnt activation has been reported in failing cardiomyocytes. Here we present single cell transcriptome profiling of hearts with inducible cardiomyocyte-specific Wnt activation (β-cat$^{\Delta ex3}$) as well as with compensatory and failing hypertrophic remodeling. We show that functional enrichment analysis points to an involvement of extracellular vesicles (EVs) related processes in hearts of β-cat$^{\Delta ex3}$ mice. A proteomic analysis of in vivo cardiac derived EVs from β-cat$^{\Delta ex3}$ hearts has identified differentially enriched proteins involving 20 S proteasome constitutes, protein quality control (PQC), chaperones and associated cardiac proteins including α-Crystallin B (CRYAB) and sarcomeric components. The hypertrophic model confirms that cardiomyocytes reacted with an acute early transcriptional upregulation of exosome biogenesis processes and chaperones transcripts including CRYAB, which is ameliorated in advanced remodeling. Finally, human induced pluripotent stem cells (iPSC)-derived cardiomyocytes subjected to pharmacological Wnt activation recapitulated the increased expression of exosomal markers, CRYAB accumulation and increased PQC signaling. These findings reveal that secretion of EVs with a proteostasis signature contributes to early patho-physiological adaptation of cardiomyocytes, which may serve as a read-out of disease progression and can be used for monitoring cellular remodeling in vivo with a possible diagnostic and prognostic role in the future.

[1] Institute of Pharmacology and Toxicology, University Medical Center Göttingen (UMG), 37075 Göttingen, Germany. [2] German Center for Cardiovascular Research (DZHK) partner site Göttingen, 37075 Göttingen, Germany. [3] Cluster of Excellence "Multiscale Bioimaging: from Molecular Machines to Networks of Excitable Cells" (MBExC), University of Göttingen, 37075 Göttingen, Germany. [4] Laboratory Animal Science Unit, Leibnitz-Institut für Primatenforschung, Deutsches Primatenzentrum GmbH, 37075 Göttingen, Germany. [5] Max-Planck-Institute for Multidisciplinary Sciences, 37075 Göttingen, Germany. [6] NGS Integrative Genomics Core Unit (NIG), University Medical Center Göttingen (UMG), 37075 Göttingen, Germany. [7] Department of Clinical Chemistry, University Medical Center Göttingen (UMG), 37075 Göttingen, Germany. [8] Bioanalytical Mass Spectrometry Group, Max Planck Institute for Multidisciplinary Sciences, 37075 Göttingen, Germany. [9] Institute for Animal Hygiene, Animal Welfare and Farm Animal Behaviour (ITTN), Stiftung Tierärztliche Hochschule Hannover, University of Veterinary Medicine, 30173 Hannover, Germany. [10] HCEMM-SU Cardiometabolic Immunology Research Group, Department of Pharmacology and Pharmacotherapy, Semmelweis University, H-1085 Budapest, Hungary. [11] Pharmahungary Group, H-1085 Budapest, Hungary. [12] Health and Medical University, D-14471 Potsdam, Germany. [13] The authors contributed equally: Eric Schoger, Federico Bleckwedel. ✉email: laura.zelarayan@med.uni-goettingen.de

Stress conditions result in cardiomyocyte growth with increased oxygen demand accompanied by an imbalanced vasculature growth ultimately resulting in deficient oxygen supply[1]. This condition affects cardiomyocytes' transcriptional profile activating adaptation processes. Due to an inefficient regenerative capacity, cardiomyocytes react to myocardial insults with limited cellular adaptation, ultimately resulting in substantial cellular loss[2]. We have previously reported that mice with inducible cardiomyocyte-specific β-catenin accumulation (β-cat$^{\Delta ex3}$) mimicked the molecular signature of hypertrophic remodeling. β-catenin is the main transcriptional transducer of the Wnt signaling pathway and was found increased in pathological heart remodeling in murine experimental models as well as in human samples[3,4], indicating that β-catenin is an evolutionary conserved hallmark of cardiac stress. Wnt signaling pathway has regenerative roles in development and disease[5]. However, activation of Wnt/β-catenin signaling specifically in cardiomyocytes resulted in molecular and phenotypic features of pathological remodeling[3]. This was characterized by a cell-autonomous effect triggered by TCF7L2/β-catenin transcriptional activation in cardiomyocytes, resulting in cell-dedifferentiation with increased cell cycle activity and heart failure[3,6,7]. These data suggested a primary regenerative function of Wnt activation, which is suppressed due to the lack of tissue plasticity in the adult heart.

Another example of Wnt regenerative functions was described upon myocardial hypoxia, which induced cardiomyocytes to release VEGF and potentially Wnt proteins that promote angiogenesis as a protective effect. However, hypoxia can enhance Wnt signaling activity by stabilizing β-catenin and altering its localization into the nucleus[8], which in turn is deleterious for cardiomyocytes. Interestingly, overexpression of β-catenin in skin fibroblasts converted them into therapeutically active cells that secrete reparative EVs in the context of myocardial infarction and Duchenne muscular dystrophy[9]. This data indicated distinct reparative and pathological roles of Wnt signaling activity in specific cell types. Today, in vivo cell specific responses mediated by Wnt induction in cardiomyocytes are not completely understood. In the present study, we aimed to further decipher these responses in a Wnt activation model specifically in cardiomyocytes as well as in hearts upon hypertrophic stress. A powerful tool that enables us to decrypt individual cellular responses within tissues is single cell RNA sequencing (SCS)[10–13]. Using this methodology, we characterized the cardiomyocytes' stress response at single cell resolution, advancing our knowledge in specific cellular responses contributing to organ remodeling. In order to achieve translational significance of our findings, we combined a cardiomyocyte-specific β-catenin gain-of-function transgenic mouse model with an experimentally induced hypertrophic remodeling in aged mice. We identified an increased unconventional vesicular-mediated secretion of proteins associated with mechanisms of stress. The vesicular-mediated secretion was validated in human induced pluripotent stem cell (iPSC) models indicating an evolutionary conservation in agreement with previously described roles of Wnt signaling in disease[14].

## Results

### Single cell transcriptomics revealed enriched EV secretion in pathological β-cat$^{\Delta ex3}$ cardiomyocytes.

Consistent with our previous observations[3], we detected significantly enriched cardiac transcripts in mouse hearts with β-catenin stabilization and Wnt target activation (β-cat$^{\Delta ex3}$) clustered to cell cycle, transcriptional and developmental processes as well as Ubiquitin-like protein (Ubl), membrane-based processes and secreted proteins (Supplementary Fig. 1A). The latter gene ontology (GO) consisted of genes involved in secreted vesicles (Supplementary Fig. 1B). In order to investigate whether secretion processes were activated by cardiomyocytes, we performed whole cell single cell transcriptome analyses from ventricular tissue of β-cat$^{\Delta ex3}$ and respective control hearts. By optimizing the isolation of heart cells, we obtained cardiomyocyte enriched cell suspensions with typical rod-shaped morphology in control hearts. Cardiomyocytes from β-cat$^{\Delta ex3}$ hearts showed the described abnormal morphology, namely larger than controls and with marked hypernucleation[3] (Supplementary Fig. 2A, B). After visualizing and confirming that single cells were dispensed, they were subjected to RNA sequencing[15–17] (Supplementary Fig. 2C, D). To gain overall insights into the cellular composition of these hearts, unsupervised clustering was applied. The entire cell population was classified into major cell types, including cardiomyocytes (CMs), endothelial cells (ECs), fibroblasts (FBs), pericytes (Per), neural-like cells (NLC) and macrophages (MΦ) based on their respective molecular markers[12,13] (Fig. 1A and Supplementary Fig. 2E). Next, we subsetted these clusters and performed pairwise differentially expressed genes (DEGs) analysis comparing control with β-cat$^{\Delta ex3}$ corresponding cell populations, focusing on cardiomyocyte and endothelial cell clusters (CM1, CM2 and EC1 and EC2, respectively). We observed differences in the cell proportions towards an increase of CM2 phenotype (Supplementary Fig. 2F). CM1 and CM2 showed common and unique up-and-down DEGs in β-cat$^{\Delta ex3}$ hearts (Supplementary Fig. 2G). A stress signature[13] was identified in CM1 and CM2 clusters of β-cat$^{\Delta ex3}$ hearts, in which CM2 showed more pronounced stress (Fig. 1B). Validating our protocol, DEGs that were previously identified by RNA-bulk sequencing analysis were also identified in cardiomyocytes derived from β-cat$^{\Delta ex3}$ hearts[3]. They included stress, cell cycle markers and Wnt targets (Acta1, Ankrd1, Myh7, Ccnd2, Nppa, Dstn, Rock2, Bambi) in CM1 and CM2 (Fig. 1C). ECs showed significantly increased cell cycle regulator Ccnd2 and developmental regulators (Anxa1, Angpt1, Dhx32, Spry1, Adam10) (Fig. 1D). In line with our RNA bulk sequencing data and the failing nature of the cardiomyocytes in β-cat$^{\Delta ex3}$ hearts, we observed gene enrichment categorizing for endosomal and vesicle-mediated transport including extracellular vesicles, multivesicular bodies (MVBs), aggresome as well as protein quality control (PQC) processes, the hypoxia pathway (HIF-1α signaling), stress response and developmental processes for upregulated genes. Distinct processes in CM1 included vascular smooth muscle contraction and proliferation, while CM2 showed the processes of spliceosome, angiogenesis and fibroblast proliferation, indicating a paracrine function of these cardiomyocytes. Common processes identified by enrichment analysis of downregulated genes for CM1 and CM2 categorized for lipid metabolism and heart development further in line with the failing condition (Supplementary Fig. 3A and Fig. 1E). Overlapping enriched processes in EC1 and EC2 from β-cat$^{\Delta ex3}$ hearts included stress, PQC processes and secretion. EC2 showed unique processes represented by angiogenesis, extracellular vesicles and cell cycle activity (Fig. 1F). Macrophages, pericytes and fibroblasts gene enrichment mainly clustered to stress pathways, secretion, PQC and vesicular transport (Supplementary Fig. 3B), all in line with the pathological cell remodeling and consequent fibrosis displayed in the β-cat$^{\Delta ex3}$ hearts[3]. These data suggested activation of the Ub-proteasome pathway, which was confirmed by higher levels of Ub-proteins in β-cat$^{\Delta ex3}$ hearts (Fig. 1G). These data distinguished specific biological activities of different cell populations in hearts with β-catenin stabilization in cardiomyocytes, which characterized pathological cardiac remodeling.

### β-cat$^{\Delta ex3}$ mice showed increased cardiac EV cargo secretion in vivo.

To investigate the role of EV-mediated processes

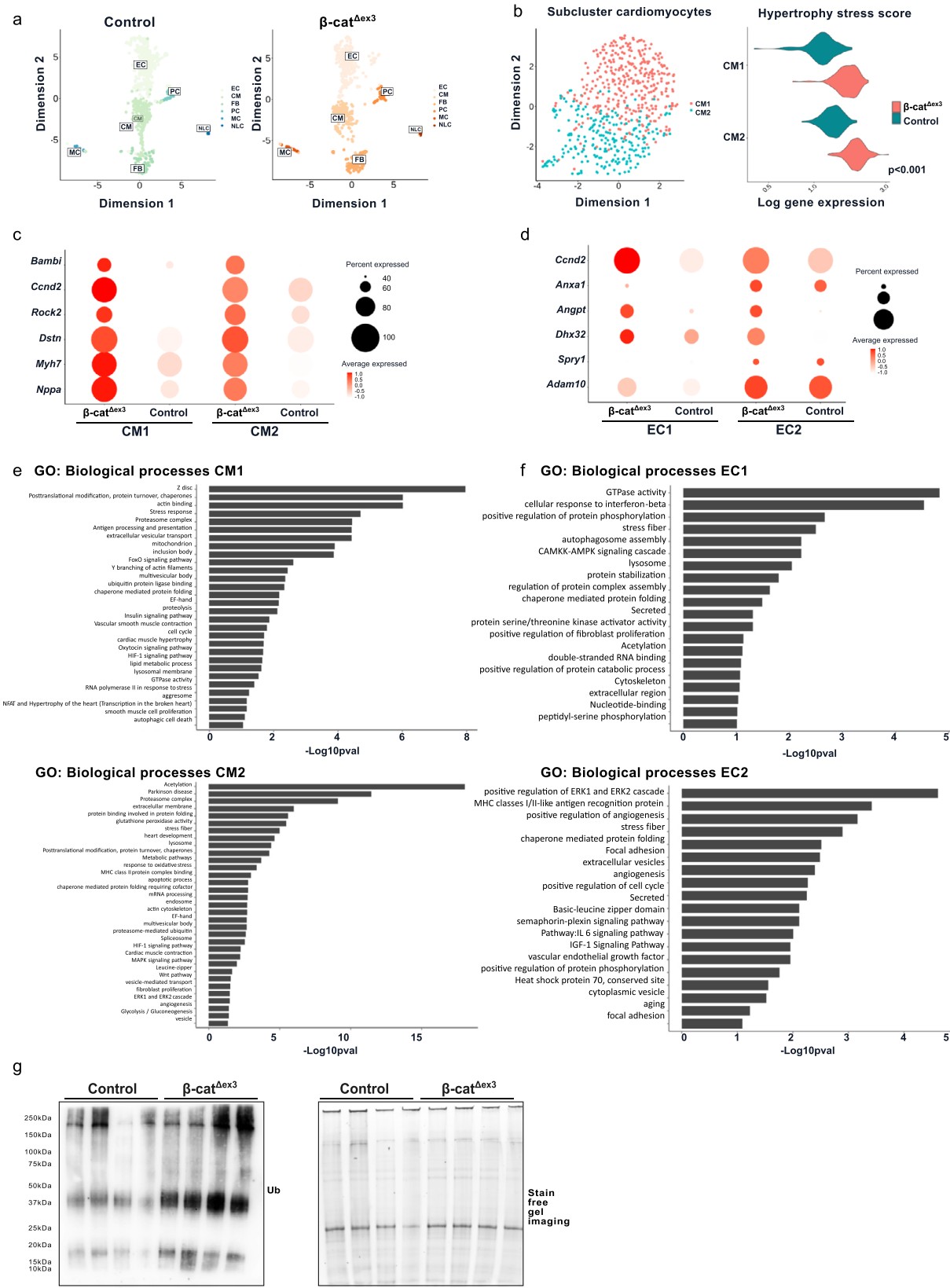

identified in the transcript enrichment, we investigated the differential cardiac EV composition in this pathophysiological context. Transferability of EV secretion studies in vitro to a disease-dependent condition is limited. Therefore, we decided to directly isolate EVs, typically present in the interstitium of cellular tissues by using solid cardiac tissue of control and β-cat$^{\Delta ex3}$ mice based on an adaption of previously described protocols[18–21]. As EV analysis ex vivo is challenging, we reasoned that our genetic model holds an advantage by focusing on changes primarily initiated from transcriptionally altered cardiomyocytes between control and β-cat$^{\Delta ex3}$ in the extracellular compartment. EVs were isolated from the concentrated heart homogenate by magnetic

**Fig. 1 Dysregulated genes and processes in cardiomyocytes and endothelial cells from β-cat$^{\Delta ex3}$ hearts. a** Representative uniform manifold approximation and projection (UMAP) plot after scRNA-seq and data integration of cardiac cells isolated from control (673 cells) or β-cat$^{\Delta ex3}$ (601 cells) hearts. Six cell clusters were identified and exclusively classified into cardiomyocytes (CM), endothelial cells (EC), fibroblast (FB), pericytes (PC), macrophage (MC) and neural-like cells (NLC) in both conditions. **b** UMAP plot of cardiomyocyte subclusters CM1, CM2 with respective hypertrophic stress scores as defined by the expression of *Acta1, Nppa, Nppb* and *Ankrd1*. **c** Dot plot depicting expression of selected cardiac stress and Wnt signaling target transcripts in cardiomyocytes from β-cat$^{\Delta ex3}$ and control hearts as indicated. **d** Selected top categories from GO biological processes enrichment of upregulated DEGs in CM1 and CM2 of β-cat$^{\Delta ex3}$ representing transcriptional responses. **e** Dot plot depicting expression of selected developmental and angiogenesis transcripts in endothelial cells (EC1 and EC2) from β-cat$^{\Delta ex3}$ and control hearts as indicated. **f** Selected top categories from GO biological processes enrichment of upregulated DEGs in EC1 and EC2 of β-cat$^{\Delta ex3}$ representing transcriptional responses in endothelial clusters. GO enrichment represents −log10 *p*-value (*p*-value <0.05) and term fusion was applied. **g** Increased Ub-protein abundance in β-cat$^{\Delta ex3}$ hearts versus control. Staining-free gel is provided as loading control.

capture beads or differential ultracentrifugation (UC) for biochemical characterization and proteomic analysis (Supplementary Fig. 4A). They were analyzed according to the minimal requirements for studies of EVs[22]. Particle size determined by nanoparticle-tracking-analysis (NTA) showed that the mean size of purified EVs was 160.0 ± 69 nm. Furthermore, cholesterol-containing vesicles were visualized using the hydrophilic fluorescence analog of cholesterol (Chol-PEG-KK114)[23] to label the P100 fraction (Fig. 2A). This fraction was further characterized by electron microscopic analysis, which showed small vesicles with uniformly round and cup-shaped morphology with a 30- to 200-nm diameter[24] (Fig. 2B). Further validating the fraction obtained by UC, Western blot analysis demonstrated the presence of EV marker CD81 in the P100 fraction, containing small EVs. In contrast, expression of Calnexin and GM130, which generally represents ER and Golgi contamination, respectively[22], was absent (Fig. 2C). Altogether, these results demonstrated that we successfully isolated EVs from heart cell homogenates providing us with, what we consider, an ideal platform for investigating in vivo produced EVs. Total protein contained in the EVs was measured, which indicated no significant differences in the EV protein content between control and β-cat$^{\Delta ex3}$ hearts (Supplementary Fig. 4B). The magnetic beads based-purified EV fraction showed the pan exosomal marker TSG101 expression and the absence of the endoplasmatic reticulum (ER) marker Calnexin, GAPDH and Vinculin, which were clearly present in cell pellets and in the P16 fraction, containing major vesicles. By loading equal amounts of protein, we observed that TGS101 expression was increased in β-cat$^{\Delta ex3}$ isolated EVs (Fig. 2D). We further aimed to gain insight into the characteristics of EVs by analyzing the protein content and employed a label-free mass spectrometry (MS) approach. From the MS data we detected and consistently quantified 573 proteins, 391 of which were significantly differentially enriched (213 increased) in preparations from β-cat$^{\Delta ex3}$ hearts (Supplementary Fig. 4C and Supplementary Data 1). 45 of the proteins were included in the Exocarta Top100 proteins found in EVs (Supplementary Data 2). Bioinformatic analysis revealed that identified proteins clustered to processes previously described in EVs[22,25,26] including extracellular exosomes (270), acetylation (265), mitochondrial (182), metabolic pathway (110), Ubl conjugation (60), Proteasome (25) and chaperone (19) (Fig. 2E). Consistent with the endosomal origin of exosomes, endosome-associated proteins were identified (Supplementary Fig. 4D). The EVs characterized in this study contained proteins found in classical small EVs or exosomes including CD81, ITGB1, ATP1B1, EHD2, and TSG101; along with cell-specific proteins including ENO1, PKM, CD47, ANXAXI and cytoskeleton as well as sarcomeric proteins. This analysis confirmed an EV nature with a major exosome representation and was therefore termed as *exosomes*. Applying the STRING functional annotation protein interaction database to all identified proteins (573), we built a protein-protein interaction (PPI) network, which clustered to

carbon metabolism, exosomes, PQC processes as well as cardiac proteins including cardiomyopathy (Supplementary Fig. 5A). They were highly enriched in proteins of the cardiovascular system (Supplementary Fig. 5B). Using the same interaction database, a PPI of the detected differentially enriched proteins (213) was performed. 202 nodes with 416 edges were identified and connected in this network with a statistical significance (*p*-value < 1.0E−16). Several hubs were identified such as proteins involved in the exosome, PQC processes, PI3K/Akt pathway as well as developmental and cardiomyopathy-related proteins. Proteins enriched in exosomes from β-cat$^{\Delta ex3}$ were associated with heart tissue, further confirming the cardiac origin of the exosomes (Supplementary Fig. 6A, B). Thus, we deciphered an enriched EV fraction containing exosomes with a cardiac proteome profile in line with the transcriptomic data.

**Cardiac exosomes from β-cat$^{\Delta ex3}$ mice showed a distinct stress-related proteome signature.** Enrichment analysis of the proteins identified in the β-cat$^{\Delta ex3}$ cardiac derived EVs included extracellular exosomes, metabolic pathways, proteasome, Z disk, cell adhesion, and cardiomyopathies among others (Fig. 3A). Cluster analysis was performed on the predicted enriched 391 proteins in exosomes from β-cat$^{\Delta ex3}$ tissue using the K-means algorithm[27]. The analysis resulted in a network with a high clustering coefficient of 0.54 and with a PPI statistical significance (*p*-value < 1.0E−16). The vast number of interactions indicated that the proteins are at least partially biologically connected. This analysis resulted in three clusters (referred to as clusters 1, 2, and 3) with 87, 52 and 63 clustered proteins, respectively. Enrichment analysis indicated that cluster 1 was enriched in metabolic proteins while clusters 2 and 3 were both enriched in PQC and exosomal processes. Additional cluster 3 proteins categorized to cardiomyopathy and sarcomeric proteins exposed to constant mechanical stress, including Desmin (DES), dystrophin muscular dystrophy (DMD); myosin binding protein C, cardiac (MYBPC3); myosin light polypeptide 2 (MYL2); Titin (TTN) and phospholamban (PLN) (Fig. 3B, C and Supplementary Fig. 7A). We further focused on cluster 3, featuring the greatest number of cardiac proteins. Using the K-means algorithm, we detected two distinguished network sub-clusters (3A and 3B), with 42 and 22-clustered proteins, respectively and with a PPI statistical significance (*p*-value < 1.0E−16). Beside extracellular exosome, GO analysis revealed that cluster 3A contained mainly proteins involved in cell adhesion, PI3K-Akt signaling pathway, laminin-complex, embryonic development and angiogenesis. Sub-cluster 3B showed a protein network categorizing for Z disk proteins, HCM, DCM and hypoxia response (Fig. 3D, E). Thus, our proteomic data showed exosomal proteins as well as context-heart-specific cargos. The sub-cluster networks of proteins may represent a pool of exosomes with different cell signatures and specific biological roles in the context of cardiac remodeling.

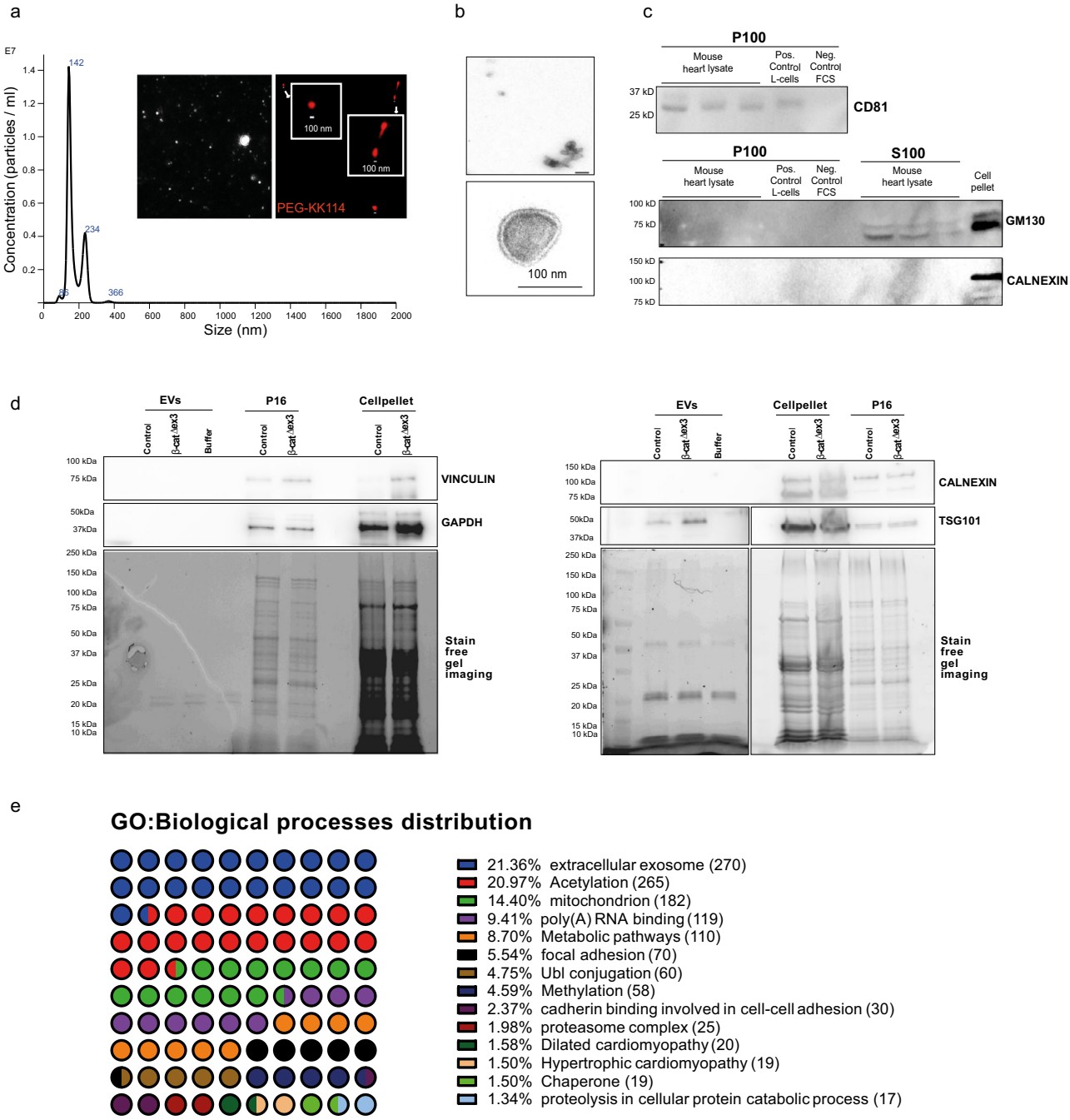

**Fig. 2 Increased cardiac EV cargo secretion in β-cat$^{Δex3}$ hearts. a** Nanoparticle tracking analyses (NTA) confirmed isolation of extracellular vesicles with a mean size of 160.0 ± 69 nm. Visualization of the purified vesicles by hydrophilic fluorescence analog of cholesterol (Chol-PEG-KK114) staining. **b** Electron microscopy confirmed the isolation of round and cup-shaped particles, typical for exosomal vesicles less than 200 nm in size. **c** Western blot showing expression of the exosome marker CD81 in the higher speed-centrifugation fraction (P100) from mouse hearts and the absence of Golgi-marker GM130 and endoplasmic reticulum marker Calnexin, which were present in the cell pellets and lower-speed centrifugation (S100). **d** Western blot showing increased exosomal marker TSG101 in EV isolated from β-cat$^{Δex3}$ hearts versus control, as well as showing absence of Calnexin, GAPDH and Vinculin expression in both conditions upon equally amount of loaded proteins as visualized by the stain free gel image. **e** GO biological processes distribution of enrichment analysis of total proteomic data (573 proteins) obtained from isolated vesicles of β-cat$^{Δex3}$ hearts and control confirming an association of vesicle contained proteins with "extracellular exosomes" ($n = 2$ (each a pool of 2-3 hearts) biological replicates and technical triplicates for NTA and MS analysis, $n = 4$, biological replicates (for Western blots). GO enrichment represents −log10 $p$-value ($p$-value <0.05) and term fusion was applied.

β-cat$^{Δex3}$ hearts purified EVs were significantly enriched for all seven α and seven β chains of the 20S proteasome components, while components of the 19S proteasome core were not enriched. We also observed enrichment of molecular chaperones, co-chaperones and PQC associated proteins previously described in secreted exosomes[28] such as HSPA70, HSP90AB1, VCP, UBE2N,

PKACA and BAG2 (shown in Fig. 3B). This went in line with the stress response of the β-cat$^{Δex3}$ hearts and elevated Ub-proteins linked to pathologic remodeling. Next, we analyzed the levels of Ub-proteins in isolated EVs and similar to the finding in the whole heart lysate, we observed increased levels of ubiquitinated proteins in EVs from β-cat$^{Δex3}$ hearts (Fig. 4A). Furthermore, the

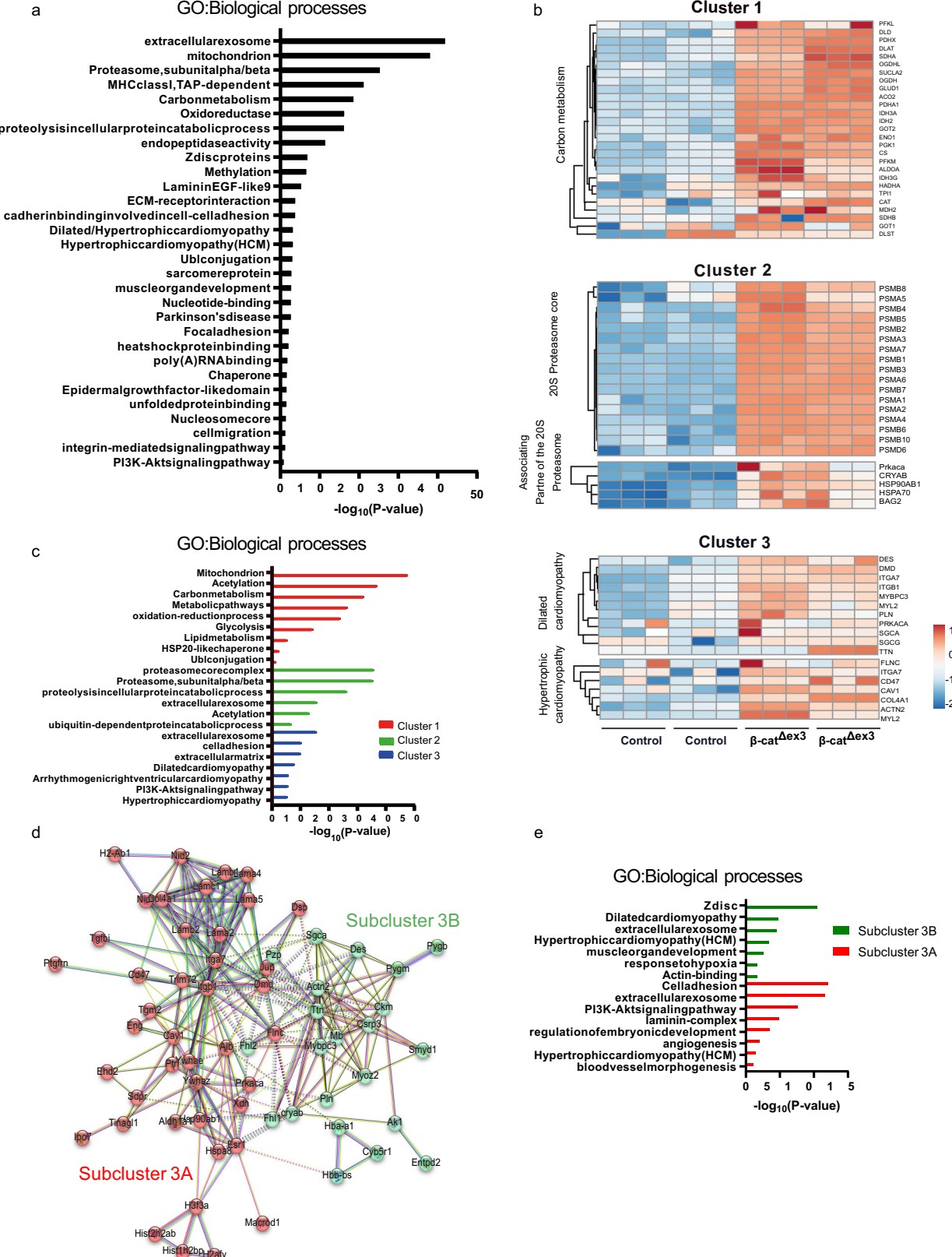

**Fig. 3 Stress-related proteome signature in cardiac exosomes from β-cat$^{\Delta ex3}$ hearts. a** GO biological processes distribution of enrichment analysis of significantly enriched proteins identified in isolated vesicles from β-cat$^{\Delta ex3}$ hearts versus control. **b** Heatmap of cluster analysis using STRING functional annotation protein interaction database on the predicted enriched loading proteins in exosomes from β-cat$^{\Delta ex3}$ tissue using the K-means algorithm[27]. **c** GO biological processes analysis of the three different clusters identified: Cluster 1 (metabolism); Cluster 2 (proteasome associated proteins) and Cluster 3 (cardiomyopathy associated proteins). **c, d** Further subclassification of the Cluster 3 in sub-clusters 3A and 3B and (**e**) associated GO biological processes. GO enrichment represents −log10 p-value (p-value <0.05) and term fusion was applied.

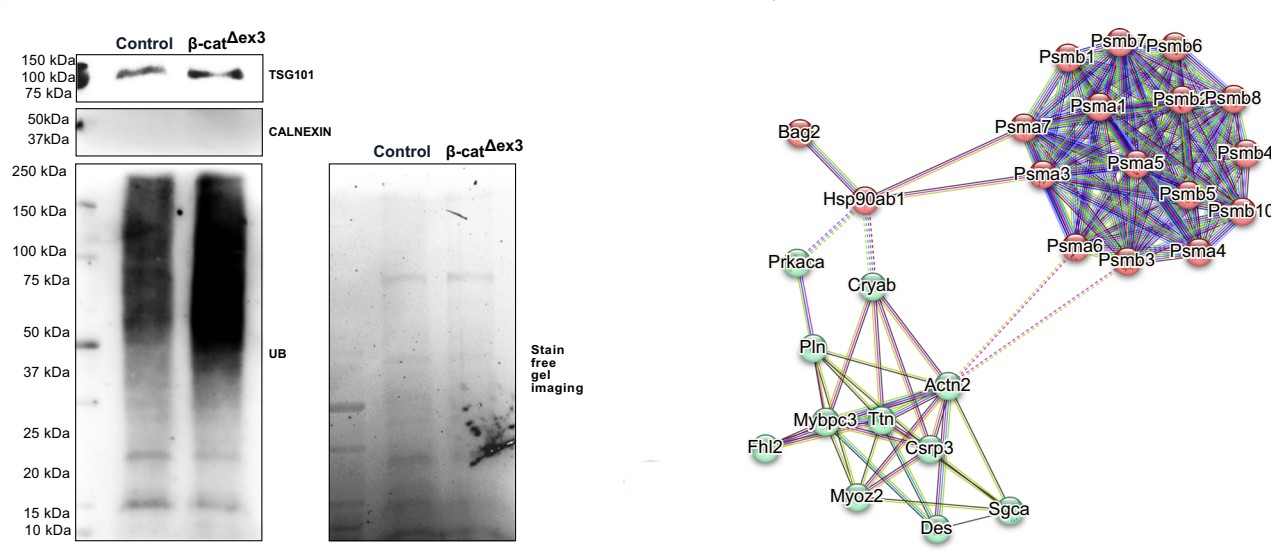

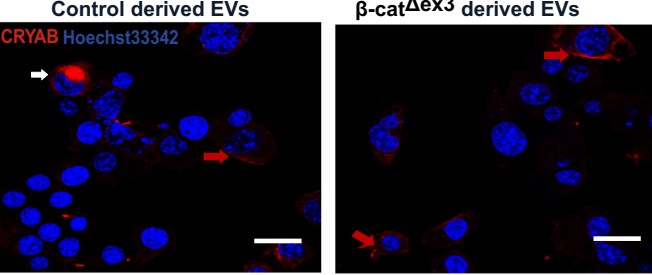

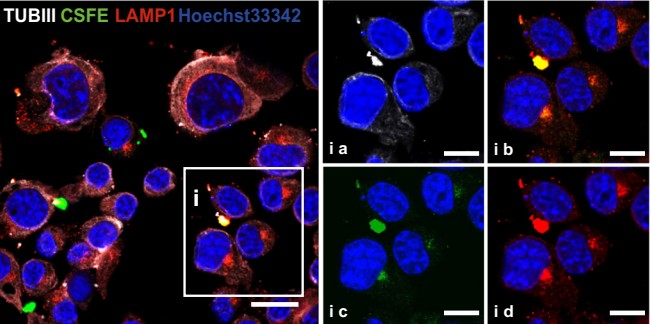

**Fig. 4 Ub-proteasomal-related proteome signature in exosomes from β-cat$^{\Delta ex3}$ cardiomyocytes. a** Increased Ub-protein abundance in β-cat$^{\Delta ex3}$ hearts versus control. Stain-free gel is provided as loading control. **b** Protein-protein interaction analysis using STRING functional annotation database on the predicted enriched proteasome, chaperones and cardiac proteins identified in β-cat$^{\Delta ex3}$ exosomes including CRYAB. **c** Representative confocal images of immunofluorescence stainings of Neuro2a cells that were exposed to β-cat$^{\Delta ex3}$ and control derived-cardiac exosomes and stained for CRYAB. Perinuclear staining (red arrows) and increased protein accumulation (white arrows) was observed in recipient cells (n = 3, technical replicates). **d** Representative confocal images of immunofluorescence staining of Neuro2a cells that were exposed to carboxyfluorescein succinimidyl ester (CSFE)-ex vivo labeled exosomes from β-cat$^{\Delta ex3}$ hearts, stained with Tubulin III (TUBIII) and LAMP1 (n = 3, technical replicates). The white box (i) indicates the region that is depicted at the right side showing the different stainings (ia-id). Hoechst33342 was used for nucleus visualization. Scale bar = 20 μm.

small heat shock protein CRYAB, that triages misfolded proteins for proteasomal degradation or repair in cardiomyopathy[29], was enriched in β-cat$^{\Delta ex3}$ hearts purified exosomes. These proteins, associated to the Z disk, showed a close interaction among the 20S core proteins, the chaperones and the cardiac proteins; suggesting a common biological process (Fig. 4B). We next incubated control and β-cat$^{\Delta ex3}$ derived-exosomes with mouse N2A neuroblastoma cells and immunolabelled for CRYAB.

Confocal microscopy analysis showed that, different to the perinuclear staining of CRYAB in both conditions (red arrows, Fig. 4C), increased accumulation of CRYAB in the membrane of cells treated with β-cat$^{Δex3}$ – cardiac derived exosomes were observed (white arrows, Fig. 4C). Next, exosomes were labeled with the membrane dye CFSE. By Z-scanning, we observed that CRYAB co-localized with CFSE-labeled exosomes attached to the cell surface (Supplementary Fig. 7B). Furthermore, cells were stained for LAMP1 and Tubulin III, which are involved in vesicle trafficking and found co-localization of these markers with the cell membrane, suggesting cellular attachment capacity (Fig. 4D).

**Differential EV-transcriptional signature characterized early and late hypertrophic cardiomyocyte populations.** We asked whether non-genetic modified cardiomyocytes upon pathological stress caused by aging and cardiac hypertrophy are characterized by similar mechanisms. Therefore, we subjected aged mice (1.5 years old) to pressure overload, induced by transaortic constriction (TAC). Hypertrophy and heart failure development was confirmed in the TAC group compared to the control sham group (Supplementary Fig. 7C). Hearts were prepared for SCS with a cardiomyocyte enriched fraction at five days (early compensatory hypertrophy (CH)) and nine weeks (late failing hypertrophy (FH)) after TAC. Unsupervised clustering revealed the presence of different cell types for all conditions (Supplementary Fig. 7D). We next focused on these cardiomyocytes and identified four distinct cardiomyocyte sub-clusters (CM1-CM4) in each condition. The cluster CM2 was mainly found in cells derived from TAC hearts at CH and FH. Only a few cells of CM2 contributed to the cardiomyocyte population in sham CH. At a later stage, more cells contributed to this cluster in the control sham hearts, which may be further associated to the aging process. Accordingly, CM2 showed an increased stress score (Fig. 5A). A pairwise DEG analysis comparing TAC with sham of the corresponding cardiomyocytes clusters revealed significant differential upregulation of hypertrophic stress and Wnt signaling markers in TAC clusters (Fig. 5B). This differential upregulation was higher at CH as compared to FH as shown by the respective overall and individual scores (Fig. 5C, D). At CH, GO analysis of upregulated DEGs revealed that common processes with highly significant enrichment shared by all the clusters included EVs, cell adhesion, PQC processes, hypertrophy and development as well as hypoxia signaling pathways. At FH, top commonly enriched processes included cytoskeletal proteins and remodeling, natriuretic peptide, cell hypertrophy and stress, blood vessel remodeling as well as alpha crystallin/Heat shock protein (Fig. 5E). Extracellular vesicular processes were enriched at FH mainly in CM2 in line with the more stress signature of this cluster (Supplementary Fig. 8A). Exosomal markers (*Cd9, Cd81, Cd63*) and *Cryab* were significantly upregulated in all CM clusters at an early stage of remodeling with a reduced differential expression at late stage remodeling (Fig. 6A, B). The exosome and hypoxic scores were significantly higher in TAC cardiomyocytes at CH and FH stages (Fig. 6C, D). Overlapping enriched processes between common DEGs from TAC and the β-cat$^{Δex3}$ cardiomyocytes at CH categorized to extracellular exosome, proteasome and proteolysis processes. At FH common DEGs categorized to development, cytoskeleton remodeling and hypertrophy indicating a hallmarks of the stress response in these cells (Supplementary Fig. 8B). Altogether, these results indicated that EV, hypoxia and PQC signaling are concomitantly and acutely activated in cardiomyocytes upon cardiac remodeling.

**Human iPSC-derived cardiomyocytes upon stress stimulus activate EV-stress signaling.** In order to investigate if the identified mechanism can be reproduced in human cells, we derived cardiomyocytes from human iPSCs and triggered stress by activating Wnt signaling to mimic our in vivo β-cat$^{Δex3}$ model. We treated the iPSC-derived cardiomyocytes with GSK-3β inhibitor (CHIR-99021), which led to accumulation of β-catenin and activation of the Wnt signaling pathway[30], or DMSO as control, for five days and medium was collected for isolation of EVs (Supplementary Fig. 9A). As a control for inducing cardiomyocyte stress, we also treated the cells with TGF-β[31]. We confirmed increased nuclear translocation of the transcriptionally active (phosphorylated (P) Ser675) β-catenin (Fig. 7A), as well as a significantly increased number of cells expressing KI67 in CHIR-treated iPSC-derived cardiomyocytes (Supplementary Fig. 9B); in line with the role of Wnt activation in cell cycle activity[3,30]. By equal protein loading of EVs isolated from CHIR-treated iPSC-derived cardiomyocytes, we observed increased levels of TSG101; which was similar to the EVs isolated from β-cat$^{Δex3}$ hearts in the isolated exosomal fraction and to a lesser extent in TGF-β treated cells. The absence of calnexin was confirmed in the exosomal fraction isolated from supernatant of iPSC-derived cardiomyocyte cultures (Fig. 7B and Supplementary Fig. 9C). Similar to the in vivo finding, we observed increased levels of ubiquitinated proteins in EVs from CHIR-treated iPSC-cardiomyocytes (Fig. 7C). Next, we performed a rescue experiment in which Wnt transcriptional activation was blocked by adding Isoquercetin, which interferes with the binding of β-catenin and the TCF/LEF transcription factors[32]. EVs isolated from these treated iPSC-cardiomyocytes showed increased expression of TSG101 upon CHIR treatment with a partial rescue upon Isoquercetin treatment. This goes in line with the increased transcriptionally active (non-phosphorylated) Ser33/37/Thr41) β-catenin in whole cell lysates of CHIR-treated iPSC-cardiomyocytes, with a non-significant reduction upon concomitant Isoquercetin treatment (Fig. 7D, E). Next, treated cells were stained with CRYAB and cardiac Troponin 2 (TNNT2) antibody and subjected to confocal microscopy analysis. This showed clear perinuclear accumulation of CRYAB in TNNT2-positive cells, which was increased as demonstrated by semiquantification of the stained area, in the CHIR-treated iPSC-derived cardiomyocytes (Fig. 7F and Extended Supplementary Fig. 9D), although total CRYAB protein levels were unchanged as shown in Supplementary Fig. 9C. This data corroborated that Wnt activation is sufficient to induce stress in human cardiomyocyte-like cells and is able to increase EV-mediated transport of stress markers of the Ub-proteasomal pathway and the CRYAB chaperone, all corresponding to our in vivo data.

**Discussion**
Active Wnt/β-catenin signaling was proven to be associated with hypertrophic heart remodeling in mouse models and human biopsies in multiple cell types including cardiomyocytes[3,4,14,33–36]. Accordingly, activation of β-catenin in cardiomyocytes resulted in molecular and phenotypic features of hypertrophic remodeling[3]. Moreover, cardiomyocyte-specific deletion of β-catenin showed an attenuated TAC-induced hypertrophic remodeling[3,37], indicating a beneficial effect of lowering Wnt activity in disease. We previously showed that β-catenin stabilization and concomitant transcriptional activation in cardiomyocytes resulted in increased transcription of developmental and cell cycle regulators. This enhanced transcriptional activation leads to cardiomyocyte polynucleation, suggesting endo-reduplication rather than newly formed myocytes, and is subsequently followed by heart failure[3], indicating that persistent reactivation of the cell cycle is not beneficial in adult cardiomyocytes. This is similar to the mechanism driven by the Hippo

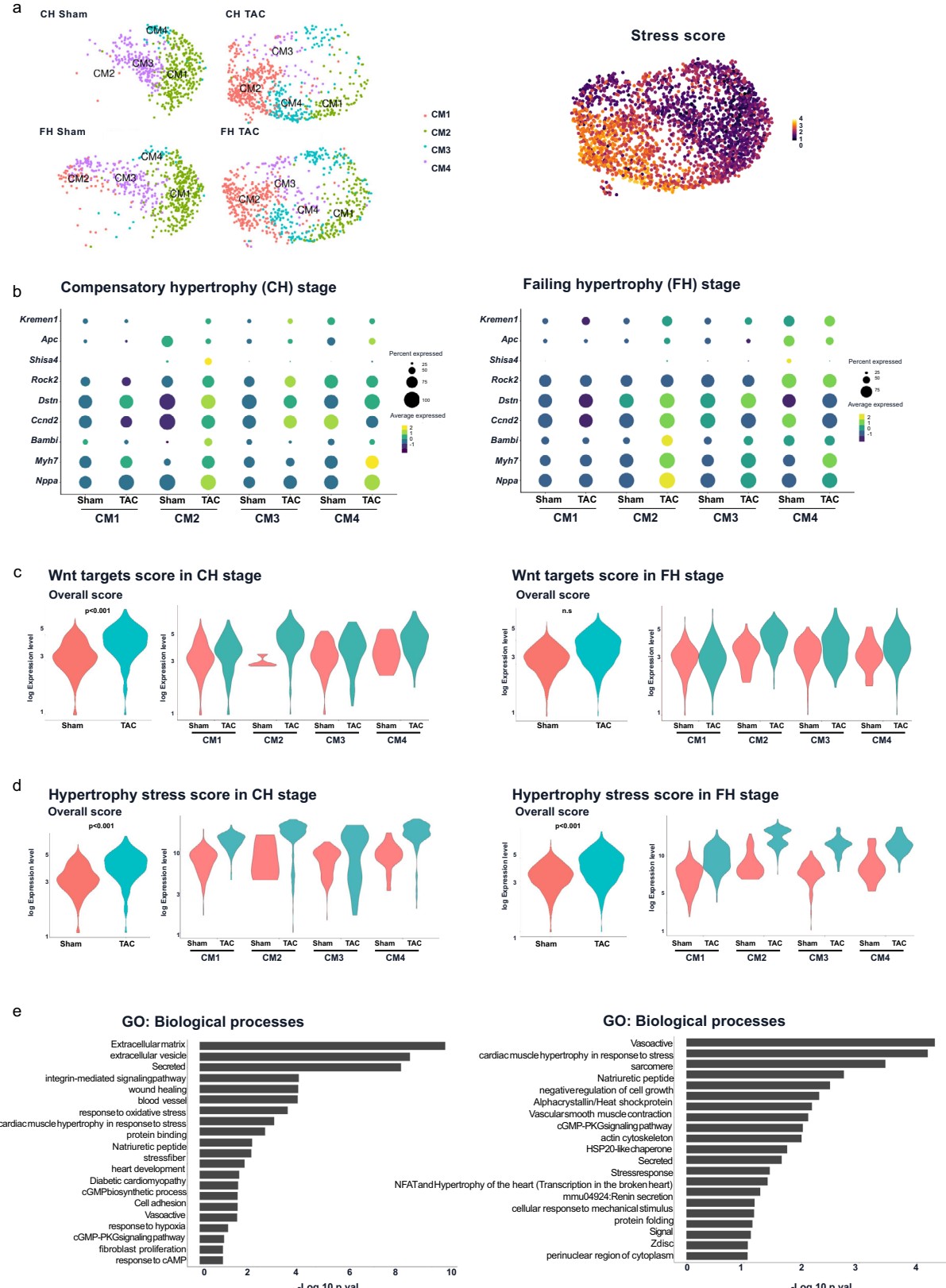

signaling pathway, which induces proliferation[38] as well as cardiomyocyte dedifferentiation and dysfunction upon long-term activation in postnatal cardiomyocytes. Thus, it seems that the activation of the Wnt cascade stimulates adaptation, dedifferentiation and proliferation of mammalian myocytes, however, the response may be determined by environmental and mechanical

cues as well as the developmental plasticity of the cardiomyocytes. Our present study showed that EV-mediated secretion was stimulated in Wnt-activated and stressed cardiomyocytes, which might be part of a compensatory adaptive response.

In line with the knowledge that EVs secretion is increased upon hypoxia[26,28,39], we observed enrichment of hypoxic signaling in

**Fig. 5 Differential EV-signaling in cardiomyocyte populations upon hypertrophic remodeling. a** UMAP plot of unbiased reclustering of the cardiomyocyte cluster identified after scRNA-seq and data integration. Cells were isolated from control (sham) and hypertrophic (transaortic constriction (TAC)) hearts at an early compensatory (CH) and late failing (FH) stages (average of 650 cells per condition and stage). Four sub-clusters were identified (CM1-CM4). Corresponding hypertrophic stress scores as defined by the expression of *Acta1*, *Nppa*, *Nppb* and *Ankrd1* are depicted for all cells.
**b** Representative dot plots showing expression levels for the stress genes *Myh7*, *Nppa*, cardiac Wnt-target genes *Dstn*, *Rock2*, *Bambi*, *Ccnd2* and Wnt inhibitors *Kremmen1*, *Shisa4*, *Apc* in clusters CM1-CM4 from sham and TAC hearts of CH (left) and FH (right) stages. Violin plots showing Wnt signaling target and stress scores in the different CM clusters for CH (**c**) and FH (**d**) stages. **e** Selected top categories from GO biological processes enrichment of overlapping DEGs obtained from sham vs TAC comparison in CH (left) and FH (right). GO enrichment represents −log10 *p*-value (*p*-value <0.05) and term fusion was applied.

those cardiomyocytes. This goes in line with both, the role of Wnt/β-catenin signaling enhancing hypoxia via HIF-1α pathway and via the hypoxia response, due to increased oxygen consumption in cardiac hypertrophy[1,40]. Further secretion functions identified in β-catenin gain-of-function cardiomyocyte clusters included angiogenesis and smooth muscle contraction. This was accompanied by endothelial cells with an active angiogenic profile and supports previous studies showing a vascular program induced by cardiomyocytes with Wnt activation in the adult heart[41]. Other studies showed that HIF1α in cardiomyocytes was inversely correlated with capillarization[7], indicating that cardiomyocytes are central for the phenotypic change of surrounding cells. We observed that stress scores were significantly increased in TAC cardiomyocytes at CH while these scores were less significant at FH, indicating a transcriptional adaptation in advanced remodeling. Another common feature of the different cardiomyocyte clusters in the heart of β-cat$^{\Delta ex3}$ mice and the early stage of induced hypertrophy was the activation of EV-related processes. Increased expression of exosomal markers was attenuated in late remodeling, strongly indicating that EV secretion processes are part of an adaptive response. Most available data on cardiac exosomes, mainly produced in cell culture experiments, indicated that their release can be enhanced by hypoxia and stress conditions[26,28,39] in line with our findings. Moreover, induced exosome production during hypoxia has cytoprotective effects in renal tubular cell injury[42]. Altogether, this provides evidence that activation of Wnt signaling along with the hypoxic response and EV-mediated secretion are part of the cardiomyocytes' transcriptional adaptive signatures upon stress.

EVs, including exosomes (or cardiosomes), were described to be released from all major heart cell types in vitro, however, it is unclear to what extent this relates to the situation in vivo[43]. In this study, we analyzed EVs in a pathophysiological in vivo context. Although no EV isolation method yet exists that can be considered a gold standard (especially from tissue) differential UC has long been regarded as a reliable technique and was employed in this study in combination with magnetic bead-based separation[44,45]. We have obtained highly consistent preparations of 30 to 200 nm diameter EVs, which expressed a set of previously described EV markers characterizing exosomes as well as showed the absence of common impurities[46,47]. In line with established guidelines[22,45], we characterized their size and morphology as well as fluorescently labeled purified EVs demonstrating colocalization with recipient cell proteins in vitro. MS analysis confirmed that the highest enriched processes, to which all proteins confidently categorized were previously described in EVs[22,25]. They include tetraspanins, integrins, growth factor receptors, cytoskeletal proteins, ESCRT-related proteins, chaperones, metabolic proteins, mitochondrial proteins and proteins involved in vesicle trafficking (annexins, major histocompatibility complex (MHC) class I and class II), as well as sarcomeric proteins[28,48]. Several of these proteins were shown increased upon hypoxia[39] and were enriched in the proteome of EVs derived from β-cat$^{\Delta ex3}$ hearts in our analysis. More importantly, cardiac proteins DES,

DMD, MYBPC3, MYL2, TTN, VPC, CD47 and PLN were enriched, strongly supporting the cardiac origin of these vesicles. Our findings therefore provide evidence for increased loading of exosomes with chaperones and sarcomeric proteins in the stressed cardiomyocytes of Wnt-activated hearts in vivo.

Significantly enriched proteins in the EVs isolated from β-cat$^{\Delta ex3}$ hearts categorized to Ub-proteasomal, metabolic and vesicular-mediated pathways along with hypertrophic cardiomyopathy. Further sub-clustering of the cardiac protein networks revealed a group of proteins categorizing to cell adhesion, PI3K-Akt signaling pathway and angiogenesis processes and another sub-cluster classifying to Z disk proteins, HCM, DCM and hypoxia response. This suggests different origins of EVs with an endothelial and a cardiomyocyte-like signature, respectively. Specifically, β-cat$^{\Delta ex3}$ hearts purified exosomes were significantly enriched for protein quality control processes including the 20S proteasome, Ub pathways, chaperones and co-chaperones previously described in secreted exosomes such as HSPA70, HSP90AB1, CRYAB, PKACA and BAG2[26,49]. Of note, PKACA was identified as a partner and regulator of murine cardiac 20S proteasomes[50], as well as CRYAB, a Z-line protein and the most abundant small HSP constitutively expressed in cardiomyocytes. CRYAB interacts with cardiac proteins including DES and TTN to prevent stress-induced aggregation or to trap prone damaged proteins upon stress[51,52]. Accordingly, we have identified loading of sarcomeric proteins prone to aggregation including DES and TTN in cardiac exosomes from β-cat$^{\Delta ex3}$ failing hearts. This is in line with the nature of Z-disk proteins that are targeted by the PQC system due to constant mechanical strain, leading to stress-induced misfolding. A mouse expressing a missense (R120G) dominant CRYAB mutant in cardiomyocytes, showed intra-sarcoplasmic amyloidosis and resulted in cardiac hypertrophy[53]. This was accompanied by an impaired proteolytic function of the Ub-proteasome system[54]. Upon acute brain ischemia-reperfusion, cardiac CRYABR120G mice showed increased inflammation with large CRYAB protein aggregates positive for ubiquitin staining in the brain. Similarly to our finding in this study showing accumulation of CRAYB in N2A cells treated with exosomes derived from β-cat$^{\Delta ex3}$ hearts, exosomes isolated from the blood of the CRYAB(R120G) mouse model induced pronounced CRYAB protein aggregation in primary neuronal cultures[51]. It was previously demonstrated that CRYAB is able to interact with the proteasome 20S Subunit Alpha 3 (PSMA3), playing important roles in the degradation of bound substrates or facilitating the degradation by the proteasome[55]. Large aggregates may remain inaccessible and impair the Ub-proteasome system activity upregulating lysosomal-autophagy-like processes[56,57]. This process may be activated in stressed cardiomyocytes, allowing CRYAB and interacting components sequestration into nascent lysosomal vesicles, which are redirected to be released as exosomes[58,59]. In our study, we identified PSMA3 amongst the cargos in the exosomes derived from β-cat$^{\Delta ex3}$ hearts as well as proteins involved in Ub signaling. Our data further suggests a role of exosome-loading with a proteostasis associated machinery at

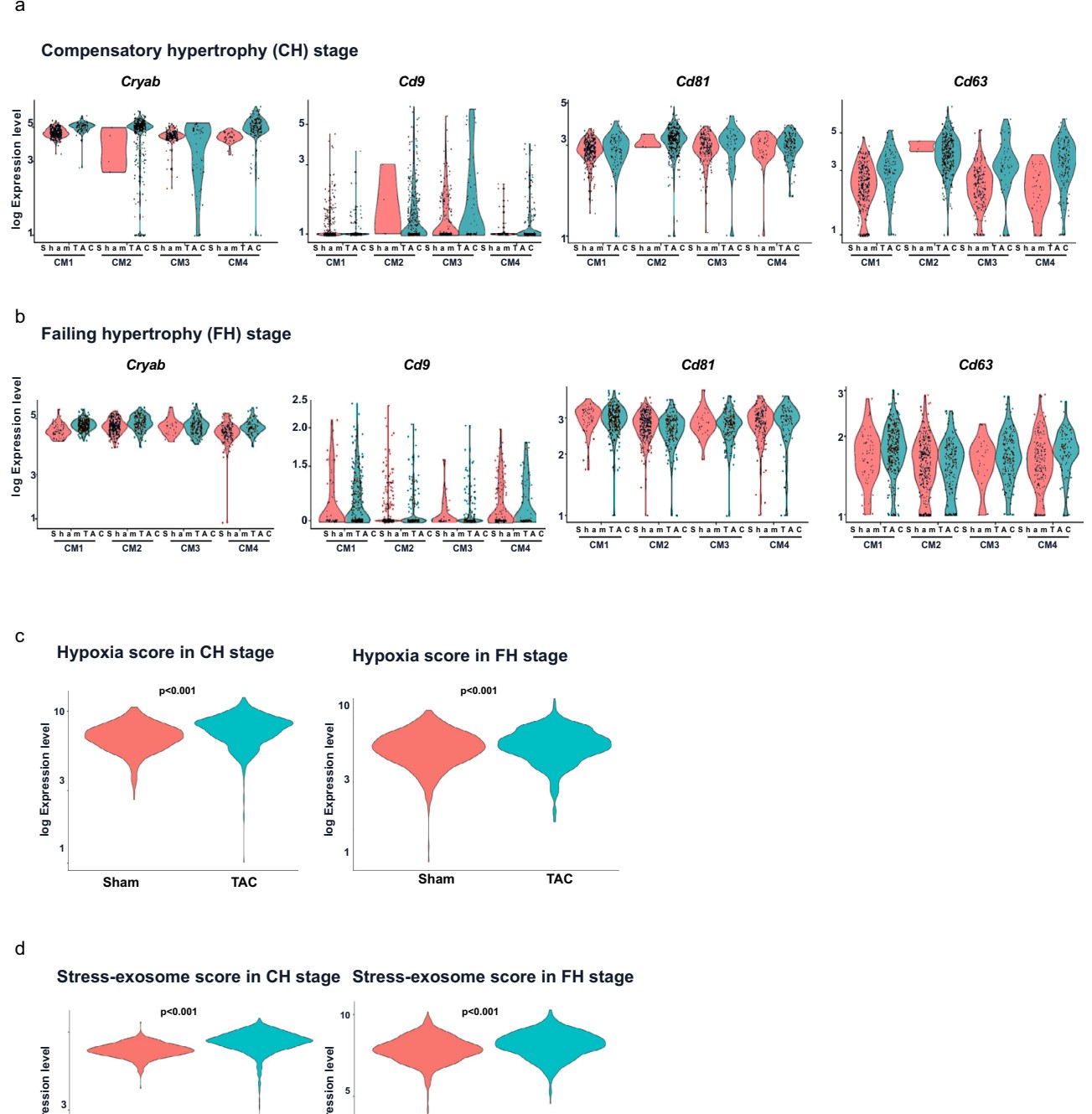

**Fig. 6 Increased exosomal and hypoxic scores in cardiomyocyte populations upon hypertrophic remodeling.** Representative violin plots showing expression levels for the cardiac chaperone *Cryab* and exosomal makers *Cd9, Cd81 and Cd63* in clusters CM1-CM4 from sham (pink) and TAC (orange) hearts of CH (**a**) and FH (**b**). Hypoxia (**c**) and exosomal scores (**d**) in sham and TAC cardiomyocytes in CH and FH.

the interface of the proteasomal and endosomal-lysosomal network upon stress conditions in cardiomyocytes. This hypothesis is based on the fact that exosome biogenesis is highly interconnected with the lysosomal degradation pathway. The endosomal pathway generates intraluminal vesicles (ILVs) that subsequently form MVBs and a hybrid organelle termed amphisome, which can either fuse to the autophagic pathway via autophagosomes or be released as exosomes by fusion with the plasma membrane[60]. Ubiquitinated proteins can serve as

markers, which are recognized and imported into a subset of ILVs that are eventually secreted as exosomes[61]. Accordingly, we have observed increased ubiquitinated proteins in EVs from Wnt activated stressed cardiomyocytes. Thus, stress conditions may favor a vesicular, unconventional secretory pathway of damaged proteins and associated machinery, possibly upon lysosomal system overload[62–66] (Fig. 8). Similarly, an increase of exosomal cargoes abundance was observed in mutated CRYAB(R120G) exosomes compared to wild-type derived exosomes along with

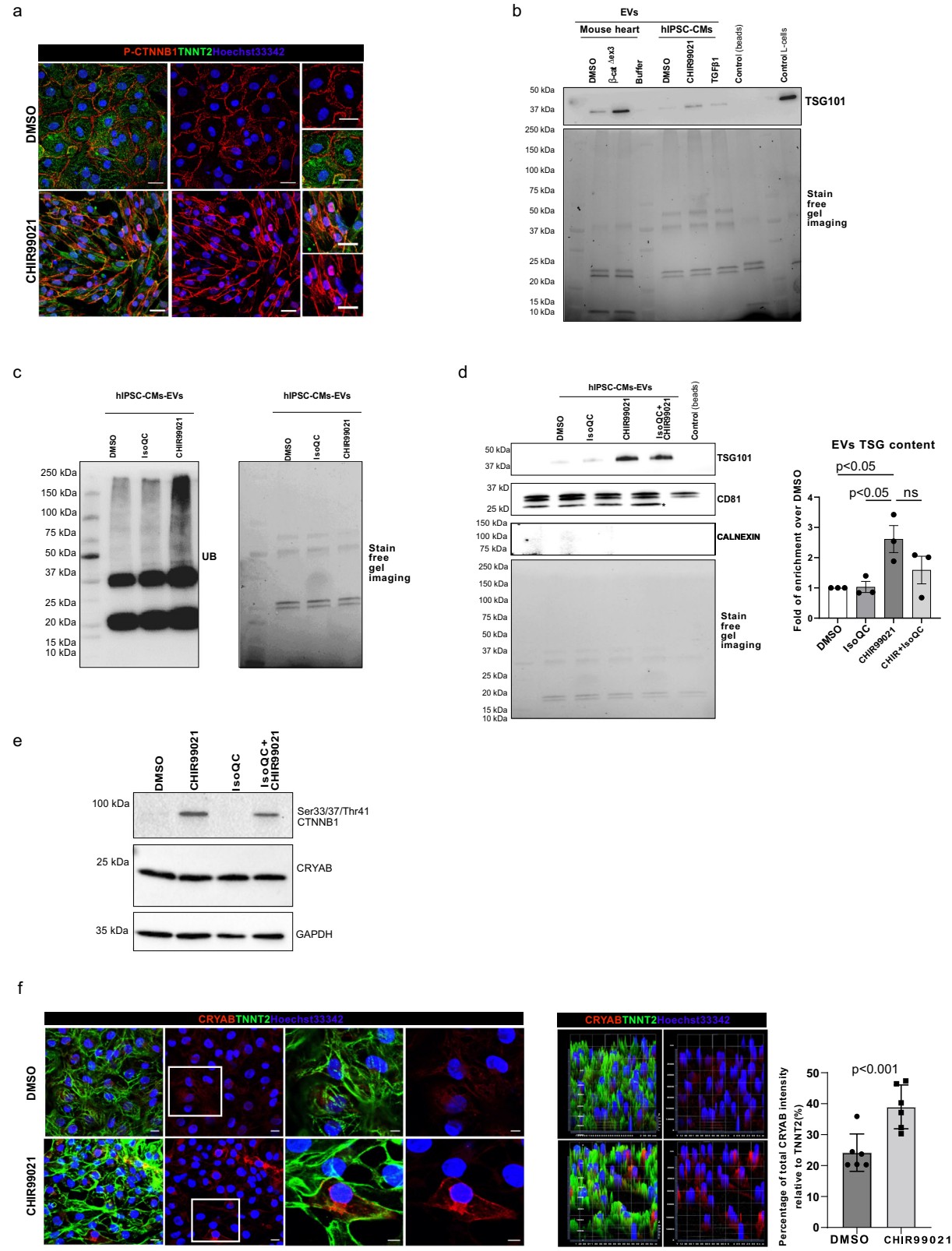

associated misfolded proteins. This may be required for the targeting of misfolded proteins to the exosome[67,68]. In line with this, significantly upregulated genes categorizing to vesicle-mediated transport, late endosome, phagosome and MVB processes as well as protein folding, chaperones, proteasome and Ubl-pathways were observed in stressed cardiomyocytes. This may represent the interconnection of cellular PQC mechanisms, whereby misfolded

proteins can either be refolded, degraded, or delivered to extracellular compartments[69]. Our data showed increased exosomal loading of proteins along with their ubiquitination upon stress caused by Wnt activation in mouse and human iPSC-derived cardiomyocytes, indicating a common EV response. Pronounced loading of EVs was also observed in iPSC-derived cardiomyocytes upon TGF-β stimulus to a lesser extent indicating a primary role

**Fig. 7 Increased exosomal cargo release in human iPSC-derived cardiomyocytes upon Wnt activation. a** Confocal immunofluorescence images showing nuclear translocation and accumulation of the transcriptionally active P-Ser675-CTNNB1 in TNNT2-positive iPSC-derived cardiomyocytes upon CHIR99021 (CHIR) treatment validating WNT/β-catenin signaling induction *versus* DMSO control-treated cells. Western blot showing increased (**b**) TSG101 levels and (**c**) Ubiquitinated protein abundance in exosomal preparations from CHIR-treated iPSC-derived cardiomyocytes *versus* control (DMSO), (TGFβ1 was included as additional stress control in **b**) upon equal amounts of loaded proteins as visualized by the stain free gel image (biological replicates n = 3 (**b**) and n = 2 (**c**) technical replicates). In **b** exosome preparation from β-cat$^{\Delta ex3}$ and control hearts were included for direct comparison. **d** Western blot and corresponding quantification showing increased TSG101 levels in exosomal preparations from CHIR-treated iPSC-derived cardiomyocytes *versus* control (DMSO) and a partial rescue upon concomitant treatment with Iso-Quercetin (Iso QC), blocking Wnt transcriptional activity. **e** Western blot showing transcriptionally active CTNNB1 and CRYAB expression in cell lysates from CHIR, Iso QC and CHIR/Iso QC treated iPSC-derived cardiomyocytes. **f** Confocal immunofluorescence images showing cytosolic accumulation of CRYAB in CHIR-treated iPSC-derived cardiomyocytes. Hoechst33342 was used for nucleus visualization and corresponding semiquantification is depicted, which is showing the intensity of CRYAB staining normalized to the total TNNT2 intensity (n = 6, technical replicates, biological duplicates). Scale bar = 20 μm. Data are shown as mean ± SEM; unpaired Student's *t* test.

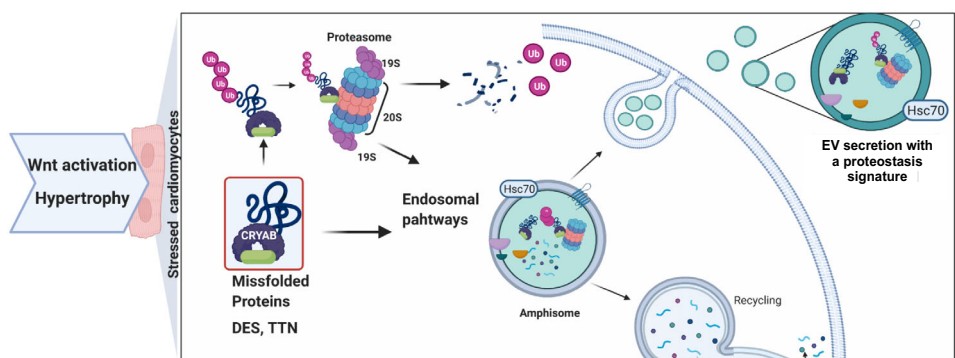

**Fig. 8 Scheme summarizing the finding in this study.** Our study revealed an increased EV-cargo secretion including Z-disk proteins prone to misfolding (DES, TNN), proteasome components and chaperone associated proteins by cardiomyocytes upon Wnt activation and pressure overload induced stress. This was accompanied by a concomitant activation of hypoxia response, which is known to activate EV-mediated processes. This response was more accentuated in early compensatory hypertrophic remodeling, suggesting their contribution to hypertrophic disease adaptation, and may overlap with the endosomal autophagic pathway at the formation of amphisomes, also activated upon stress. It is tempting to speculate that increased cellular stress induces protein misfolding, which triggers an excess of ubiquitination and CRYAB-mediated processes activation and the amphisomes are redirected to the exosomal release. Created with BioRender.com.

of hypertrophic stress in this process. Rescuing Wnt activation resulted in a mild reduction of the increased exosome cargoes, indicating that Wnt activation, and more generally hypertrophic stress, is responsible for the described phenotype.

In summary, this study characterized the proteomic cargoes of cardiac-derived EVs in an in vivo context of failing cardiomyocytes. Here, we complemented the analysis of Wnt-activated cardiomyocyte in vivo, in order to precisely study the mechanisms driven by Wnt signaling, with a hypertrophic stress model, as a translational approach. Furthermore, we extended our analysis on human-derived cardiomyocytes to study the evolutionary conservation of the identified mechanism. It revealed an increased EV-mediated process associated with proteins of the Ub-proteasome system released by cardiomyocytes, suggesting their contribution to hypertrophic disease adaptation. It is tempting to speculate that the increased cellular stress induces protein misfolding, leading to an excess of ubiquitination and activation of chaperone-mediated processes, which are redirected as exosomal release. This may represent an additional "back-up" system, by which cells alleviate the accumulation of intracellular aggregate-prone or misfolded proteins. As such, this could ultimately limit the pathogenesis and the onset of misfolding associated disease mechanisms, and in perspective, represent a potential target to develop novel therapeutic intervention[70]. It may be interesting to identify the presence of these EVs in the circulating blood as previously demonstrated in cardiomyocyte-derived EVs, which were present in the circulation and reflecting cardiac injury that can serve as biomarker of cardiac injury[71,72]. The EVs protein cargo composition associated with proteostasis

functions of the cardiomyocytes make them attractive potential biomarkers to detect cardiac health[73]. Finally, since exosomes are messengers from one cell to another, secretion of unwanted material to the extracellular environment may have an impact, beneficial or detrimental, on neighboring cells[61] and therefore deserves further investigation.

## Methods

**Animals**. Both male and female mice were included in the study and were kept in a 12 h day/night cycle with *ad libitum* feeding. The generation of the heart-specific β-catenin gain-of-function (β-Cat$^{\Delta ex3}$) mouse model was previously described[3] (C57BL/6N background). Briefly, for transgene induction in β-Cat$^{\Delta ex3}$, heart-specific expression of the Cre recombinase under control of the *Myh6* promoter was activated by administration of Tamoxifen (T5648, 30 mg/kg body weight/day; Sigma–Aldrich) intraperitoneal for 3 days. Wild-type β-catenin and Cre recombinase positive littermates were used as control for β-Cat$^{\Delta ex3}$ mice. Genotyping was performed as previously described[3] with the following primers: Fwd: 5' GCTGCTGTGACACCGCTGCGTGGAC 3' Rev: 5' CACGTGTGGCAAGTTCC GCGTCATCC 3' using the REDExtract-N-Amp PCR ReadyMix (Sigma-Aldrich). Transverse aortic constriction (TAC) was performed in 17,5 months-old (average) mice as previously described[3,7]. The intervention was performed by tying a braided 5-0 polyviolene suture (Hugo Sachs Elektronik) ligature around the transversal aorta and a blunted 26-gauge needle and subsequent removal of the needle. For sham controls the suture was not tied. To determine the level of pressure overload by aortic ligation, a high frequency Doppler probe was used to measure the ratio between blood flow velocities in right and left carotid arteries. TAC mice with blood flow gradient <60% were excluded. Echocardiography analysis were done at different time points and for that, mice were anesthetized with 2% isoflurane inhalation and ventricular measurements were performed with a VisualSonics Vevo 2100 Imaging System equipped with a MS400, 30 MHz MicroScan transducer. The observer was unaware of the genotypes and treatments. All procedures were performed by the SFB 1002 service unit (S01 Disease Models). All animal experiments were approved by the animal research review board Lower Saxonian State Office for Consumer Protection and Food Safety (Niedersächsisches

Landesamt für Verbraucherschutz und Lebensmittelsicherheit (LAVES)) (AZ-G 20.3434).

**Single cell sample preparation**. Bulk RNAseq was previously published[3]. For SCS, a modified Langendorff perfusion was performed. Mice were anesthetized with isoflurane. Hearts were dissected and canulated via the ascending aorta with an 18 G blunt-end needle and retrogredely perfused with Perfusion Buffer (150 mmol/L NaCl, 5 mmol/L HEPES, 5.4 mmol/L KCl, 10 mmol/L glucose, 2 mmol/L Na-pyruvate, 1.2 mmol/L $MgCl_2 \cdot 6\ H_2O$, 10 mmol/L taurine, 12.38 mmol/L 2,3-butanedione monoxime, pH 7.35). Hearts were enzymatically digested with Digestion Buffer (210 µg/mL Liberase DH (Roche), 25 µmol/L $CaCl_2$ in Perfusion Buffer) for 10 min at 37 °C, minced into tissue pieces, homogenized by pipetting with a 10 ml serological pipette to yield a single cell suspension of rod-shaped cells. Debris was removed by washing with Exchange Buffer (0.5 w/v % bovine serum albumin, 200 µmol/L $CaCl_2$ in Perfusion Buffer) and gravitational settling of tissue pieces for 1 min.

**Single cell library preparation**. Samples were prepared, processed and analyzed for single cell transcriptomics at NIG, Institute of Human Genetics, University Medical Center Göttingen. Briefly, cells were distributed on 5,184 nanowell chips ICELL8 250v Chip (ICELL8 System, Takara Bio). Single alive cells were identified using Hoechst 33342 and propidium iodide staining (NucBlue Cell Stain Reagent, Thermo Fisher Scientific) and the CellSelect Software (Takara Bio). Complementary DNA synthesis was performed by oligo-dT priming in a one-step RT-PCR reaction. P5 indexing primers, Terra Polymerase and Reaction Buffer were added for library preparation. Transposase enzyme and reaction buffer (Tn5 mixture) were added to each well. P7indexing primers were dispensed to wells. Final libraries were amplified and pooled as they are extracted from the chip. Pooled libraries were purified and size selected with Agencourt AMPureXP magnetic beads (Beckman Coulter) to obtain an average library size of 500 bp. Libraries were sequenced with a HiSeq4000 (Illumina) to obtain on average ~$3 \cdot 10^5$ reads per cell (single-end, 50 bp).

**SCS data analysis**. Raw sequencing files (bcl-files) were converted into a single fastq file using Illumina bcl2fastq software (v2.20.0.422) for each platform. Each fastq file was demultiplexed and analyzed using the Cogent NGS analysis pipeline (CogentAP) from Takara Bio (v1.0). In brief, cogent demux wrapper function was used to allocate the reads to the cells based on the cell barcodes provided in the well-list files. Subsequently, cogent analyze wrapper function performed read trimming with cutadapt[74](version 3.2), genome alignment to *Mus musculus* genome GRCm38 using STAR[75] (version 2.7.7a), read counting for exonic, genomic and mitochondrial regions using featureCounts[76] (version 2.0.1) and utilizing *Mus musculus* gene annotation version 102 from ENSEMBL and generating a gene matrix with the number of reads expressed for each cell in each gene. Raw gene matrices underwent quality control (QC) filtering for cells and genes using the following parameters: (a) for cells, only those with at least 2500 genes and less than 30 % of mitochondrial reads, and (b) for genes, only those containing at least 100 reads mapped to them from at least 3 different cells. The bioinformatics analysis was performed using the Seurat package (version 4.1.2)[77]. Expression matrices were split by Chip and, after normalization by 'NormalizeData' using default settings and selection of the 2000 most variable features using 'FidVariableFeatures'. Integration of the datasets was performed by the standard parameters of 'IntegrateData'. Data were scaled and centered by 'ScaleData', regressing out the variability caused by the mitochondrial percentage and the Depth of the sequencing. Dimensions were reduced using 'RunPCA'. Afterward, 'FindNeighptors' (dimension parameter = 1:9) and 'FindClusters' (granularity parameter = 0.3) were run and the clusters were visualized after UMAP reduction was calculated by using Louvian algorithm. Gene ontology (GO) analyses were performed using default parameters and stringency in 'ClueGO' (Cytoscape)[78] and DAVIS. The significant gene ontologies (GO) are shown with $p < 0.05$. The cardiomyocyte stress, hypoxia and EV scores were calculated by the expression of established markers per cell[13]. Ggplot2[79], Dplyr[80] and EnhancedVolcano packages were used for data visualization[81].

**Exosome isolation**. For proteomic analysis and cell treatment, cardiac EV isolation from mouse hearts was performed with Langendorff perfusion as described above. The hearts were disconnected from the Langendorff apparatus, atria were separated and ventricles were minced in a petri dish with 2.5 mL Exchange buffer. The homogenate was transferred to a Falcon tube and the Petri dish was rinsed with an additional 5 mL of Exchange buffer, which was also transferred to the Falcon tube. Non-digested heart tissue was allowed to sediment for 1 min. Supernatant was filtered through a 40 µm cell strainer. All steps took place at 4 °C. Exosomes were isolated using subsequent steps of centrifugation. First, samples were centrifuged at $2000 \times g$ for 10 min to pellet remaining cells, cell debris and large apoptotic bodies. In the next step, supernatants were centrifuged at $14,000 \times g$ for 35 min to remove microvesicles and other large extracellular vesicles. Supernatants were transferred to ultracentrifugation tubes and centrifuged at $100,000 \times g$ for 2 h at 4 °C. Pellets were then washed with PBS and recentrifuged at $100,000 \times g$ for 1,5 h. Supernatants were kept (S100) and pellets resuspended (P100) in 50–100 µl of PBS or RIPA

buffer for Western blot and stored at −20 °C until further analysis. For biochemical characterization, cardiac EV isolation from mouse hearts and supernatants obtained from human iPSC-derived cardiomyocytes was performed by positive selection Magnetic Activated Cell Sorting (MACS). The mice´s hearts were removed and placed in a 35 mm PBS Petri dish. Then, the hearts were sliced using a sterile scalpel and dissociated until having a homogenous size of approximately $2 \times 2 \times 2$ mm. After that, the pieces were transferred into a 6 well plate and immediately enzymatically digested using Liberase (1 mg/mL) for 30 min at 37 °C. Subsequently, the lysate was transferred to a fresh 15 mL tube and homogenized using a 1 mL. pipette. Tissue was allowed to settle for a minute and next the supernatant was removed and filtered using a 40 µm cell strainer into a 50 mL tube. Following this, removal of cellular debris by serial centrifugation ($2000 \times g$ and $10,000 \times g$) digested ventricles or supernatants were incubated for 1 h with Isolation kit MicroBeads (Mouse Exosome Isolation kit Pan, Miltenyi Biotec). After incubation, labeled EVs were loaded on a magnetic column, washed to remove unlabeled vesicles and eluted with PBS for protein quantification and Western blot analysis.

**Transmission electron microscope (TEM)**. TEM grids (copper, 150 hexagonal mesh, Science Services, Munich, Germany) which were coated with a formvar film were put on top of 10 µl droplets of the exosome fraction and incubated for 10 min. Then, the grids were washed 5 times with PBS followed by incubations on droplets of water. For contrasting, the grids were stained for 5 min on droplets of urany-lacetate-oxalate, followed by 5 min incubation on droplets of a 1:9 dilution of 4% uranylacetate in 2% methylcellulose. These solutions were prepared as described[82]. After blotting the methylcellulose from the grids using filter paper and drying of the methylcellulose film, samples were imaged with a LEO912 transmission electron microscope (Carl Zeiss Microscopy, Oberkochen, Germany) and images were taken using an on-axis 2k CCD camera (TRS, Moorenweis, Germany).

**Nanoparticle tracking analysis (NTA)**. The size distribution of the serum exosomes was analyzed using a Malvern Panalytical NS300 instrument equipped with nanoparticle particle tracking software (Version NTA 2.3 Analytical Software). According to the manufacturer's recommendation, the samples were illuminated by the laser (Blue 488) and the movement of nanoparticles due to Brownian motion was recorded for 60 s at a mean frame rate of 20 frames per second. Each process was repeated three times. sEV particles were diluted in PBS (1:100). Camera level 14, screen gain 10.8, detection threshold 5. For each sample, a total of three videos of 30–60 s was measured. The videos were analyzed by the NanoSight NTA and the particles' concentration, size distribution and the general mean and mode of the samples were obtained.

**Western blot analysis (WB)**. Proteins were extracted from EV preparation and tissue by addition of lysis buffer (150 mmol/L sodium chloride, 1.0% NP-40 or Triton X-100, 0.5% sodium deoxycholate, 0.1% SDS (sodium dodecyl sulfate), 50 mmol/L Tris) or 10 mmol/L Tris/HCl pH 7.5; 150 mol/L NaCl; 0.5 mmol/L EDTA; 0.5% NP-40) containing proteases and phosphatases inhibitor cocktail (Roche). After clarification ($12,000 \times g$ at 4 °C, 20 min), protein concentration was quantified with Bradford assay and the different samples were resuspended in sample buffer. Bio-Rad system was used to run the blots, at constant 120 V, followed by semi-dry transfer at constant 140 mA. After blocking (3% BSA in TBST) the membranes were incubated with primary antibodies overnight (Supplementary Data 3). Membranes were incubated in secondary antibody for 1–2 h at room temperature (anti-mouse IgG (H + L) HRP conjugate (Bio-Rad) 1:1,000), developed (ECL Amersham) and imaged using ChemiDoc Touch Imaging System (Bio-Rad). All unedited blots are provided in Supplementary Fig. 10.

**Liquid chromatography coupled to tandem mass spectrometry (LC-MS/MS)**. Samples were reconstituted in 1× NuPAGE LDS Sample Buffer (Invitrogen) and applied to 4–12% NuPAGE Novex Bis-Tris Minigels (Invitrogen). Samples were run 1 cardiomyocyte into the gel for purification and stained with Coomassie Blue for visualization purposes. After washing, gel slices were reduced with dithiothreitol (DTT), alkylated with 2-iodoacetamide and digested with trypsin overnight. The resulting peptide mixtures were then extracted, dried in a SpeedVac, reconstituted in 2% acetonitrile/0.1% formic acid/ (v:v) and prepared for nanoLC-MS/MS[83]. All samples were spiked with a synthetic peptide standard used for retention time alignment (iRT Standard, Schlieren, Schweiz). Protein digests were analyzed on a nanoflow chromatography system (Eksigent nanoLC425) hyphenated to a hybrid triple quadrupole-TOF mass spectrometer (TripleTOF 5600+) equipped with a Nanospray III ion source (Ionspray Voltage 2400 V, Interface Heater Temperature 150 °C, Sheath Gas Setting 12) and controlled by Analyst TF 1.7.1 software build 1163 (all AB Sciex, Darmstadt, Germany). In brief, peptides were dissolved in loading buffer (2% acetonitrile, 0.1% formic acid in water) to a concentration of 0.3 µg/µl. For each analysis, 1.5 µg of digested protein were enriched on a pre-column (0.18 mm ID × 20 mm, Symmetry C18, 5 µm, Waters, Milford/MA, U.S.A) and separated on an analytical RP-C18 column (0.075 mm ID × 250 mm, HSS T3, 1.8 µm, Waters) using a 90 min linear gradient of 5–35% acetonitrile/0.1% formic acid (v:v) at 300 nl min-1. Qualitative LC/MS/MS analysis was performed using a Top25 data-dependent acquisition (DDA) method with an MS survey scan of *m/z*

350–1250 accumulated for 350 ms at a resolution of 30,000 full width at half maximum (FWHM). MS/MS scans of $m/z$ 180–1,600 were accumulated for 100 ms at a resolution of 17,500 FWHM and a precursor isolation width of 0.7 FWHM, resulting in a total cycle time of 2.9 s. Precursors above a threshold MS intensity of 125 cps with charge states $2^+$, $3^+$, and $4^+$ were selected for MS/MS, the dynamic exclusion time was set to 30 s. MS/MS activation was achieved by CID using nitrogen as a collision gas and the manufacturer's default rolling collision energy settings. Two technical replicates per sample were analyzed to construct a spectral library. For quantitative analysis, MS/MS data were acquired using data-independent acquisition (DIA) with 65 variable-size windows[84] across the 400–1050 $m/z$ range. Fragments were produced using rolling collision energy settings for charge state $2^+$, and were acquired over an $m/z$ range of 350–1400 for 40 ms per segment. Including a 100 ms survey scan, this resulted in an overall cycle time of 2.75 s. 3 technical replicates were acquired for each sample. Protein identification was achieved using ProteinPilot Software version 5.0 build 4769 (AB Sciex) at "thorough" settings. The combined qualitative analyses were searched against the UniProtKB mouse reference proteome (revision 12-2017, 60717 entries) augmented with a set of 52 known common laboratory contaminants to identify proteins at a False Discovery Rate (FDR) of 1%. Spectral library generation and SWATH peak extraction were achieved in PeakView Software version 2.1 build 11041 (AB Sciex) using the SWATH quantitation microApp version 2.0 build 2003. Following retention time correction using the iRT standard, peak areas were extracted using information from the MS/MS library at an FDR of 1%[85]. The resulting peak areas were then summed to peptide and finally protein area values per injection, which were used for further statistical analysis in Perseus 1.5.6.0 software (Max Planck Institute for Biochemistry, Martinsried, Germany). STRING (Search Tool for the Retrieval of Interacting Genes/Proteins) (http://string-db.org/) was used for network analysis. Gene ontology (GO) analyses were performed using default parameters and stringency in 'ClueGO' (Cytoscape)[78] and DAVIS.

**Exosome labeling**. For immunofluorescence analysis, the exosome pellet was resuspended in PBS and stained with Chol-PEG-KK114 (kindly provided by Volker Westphal and Stefan Hell, Max-Planck-Institute for Biophysical Chemistry)[23,86]. Exosomes isolated from one heart were incubated with Chol-PEG-KK114, diluted 1:100 from a 100 µmol/L stock solution, for 30 min at RT, washed with PBS twice by ultracentrifugation, fixed and placed on coverslips for imaging in Zeiss LSM 710 NLO confocal microscope. For EV protein labeling, 250 µL diluted EV samples (containing 2 µg Exosomes) were incubated with 5 µM Carboxy-fluorescein succinimidyl ester (CFSE) (Thermo Fischer Scientific) in microtiter wells for 45 min at 37 °C. Cells were incubated with labeled exosome dilutions in a Yokogawa high-content screening system and observed for 3 h (37 °C, 7% CO2). The pinhole was set to 50 µm.

**Human induced pluripotent stem cell (iPSC)-derived cardiomyocytes**. Human iPSC LiPSC-GR1.1 (TC1133 or RUCDRi002-A)[87] were cultured on Growth Factor Reduced Matrigel (BD Biosciences) in StemMACS iPSC-Brew XF, human (Milteny Biotec) differentiated by WNT modulation into iPSC-derived cardiomyocytes (Base Medium: RPMI 1640+ GlutaMAX, 2% B27 supplement (Thermo Fisher Scientific), 200 µmol/L L-ascorbic acid, 1 mmol/L Na-pyruvate, 100 U/mL penicillin, 100 µg/mL streptomycin (Gibco)) by mesoderm induction with 9 ng/mL Activin A (Bio-Techne), 1 µmol/L CHIR99021 (Merck Chemicals GmbH), 5 ng/mL BMP4 (Bio-Techne) and 5 ng/mL FGF (PeproTech) for three days followed by cardiac differentiation by WNT inhibition with 5 µmol/L IWP4 (ReproCELL Europe Ltd.) for 9 days. Cardiomyocytes were metabolically selected (RPMI 1640 without glucose, 2.2 mmol/L Na-lactate, 100 µmol/L β-mercaptoethanol (Gibco), 100 U/mL penicillin, 100 µg/mL streptomycin) for a total period of approximately four weeks before treatment. iPSC-derived cardiomyocytes were treated with 10 µmol/L CHIR99021 (Merck Chemicals GmbH) alone or in combination with 150 µmol/L Isoquercetin (Selleck Chemicals GmbH), 10 pmol/L TGFB1 (Pepro-tech) or DMSO as control prepared in RPMI1640 without glucose, 4 mmol/L Na-pyruvate, 200 µmol/L L-ascorbic acid, 2% B27 supplement minus insulin, 100 U/mL penicillin, 100 µg/mL streptomycin for five days in a 12-well plate. Supernatant was collected every day and pooled for EV isolation. Cells that were cultured on Matrigel (Corning) coated coverslips were fixed and used for immunofluorescence analysis.

**Immunohistochemistry**. For immunofluorescence, Neuro2A cells (ATCC CCL-131) and iPSC-derived cardiomyocytes plated on Matrigel (Corning) coated glass coverslips were fixed with 4% PFA, followed by PBS washing and permeabilization with 0.2% BSA and 0.3% Triton X-100 in PBS for 10 min. Cells were then blocked with 5% BSA and 0.1% Triton X-100 at RT. Primary and secondary antibodies (Supplementary Data 3) were diluted in 2% BSA and 0.1% Triton in PBS. Coverslips were mounted with ProLong Gold medium containing DAPI (Invitrogen). Murine heart tissues were dissected, rinsed in PBS, fixed in 4% PFA overnight at 4 °C, embedded in paraffin and sectioned at 3 µm thickness using a Leica microtome. Paraffin-embedded sections were de-paraffinized and rehydrated, and antigen was unmasked by microwaving sections for 10 min in 10 mmol/L sodium citrate buffer, pH 6. Sections were blocked at RT for 1 h with 5% BSA and 0.1%

Triton X-100 in PBS. For immunostaining, primary antibodies (Supplementary Data 3) were incubated overnight at 4 °C. Next, sections were washed in PBS and incubated with the corresponding secondary IgG-Alexa488, 594 or 633 antibodies (1:200, Molecular Probes). Slides were imaged using a Zeiss LSM 710 NLO confocal microscope. Fluorescence intensity was analyzed with ZEN lite 2012 software.

**Statistics and reproducibility**. G-Power3.1 was used to determine the sample size for animal studies. For single cell analysis, a priori: compute required sample size, was used. Sample size per group = 105 (Effect size d = 0.5, power = 0.95, comparison of two independent groups). For mice interventions, sample size per group=3 (Effect size d = 0.9, power = 0.85, comparison of two independent groups). Statistical analyses were performed using GraphPad PRISM 7. Two-tailed unpaired Student's $t$ tests were conducted on two group comparisons. Pearson's correlation was performed for correlation plots. Statistical significance was assumed when $p < 0.05$ (*). Specific statistics applied for proteomic and transcriptomic analysis are mentioned in the corresponding sections.

**Reporting summary**. Further information on research design is available in the Nature Portfolio Reporting Summary linked to this article.

## Data availability

The mass spectrometry proteomic dataset generated during the current study is available in the ProteomeXchange Consortium via the PRIDE[88] partner repository with the dataset identifier PXD031113, the bulk RNAseq data is available at NCBI GEO under accession GSE97763, the raw and normalized Single Cell RNAseq data is available at https://owncloud.gwdg.de/index.php/s/NeSJGuf2o2yc2qn. Reporting of the EV protocol is available at EV-TRAC[89] including the MISEV2018[22] (ID: EV220418). The uncropped Western blots in Supplementary Figure 10.

## Code availability

The full code used during the current study is available at https://github.com/fblec/Stressed_CMs_EVs.

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

## Acknowledgements
We thank Tobias Weber and Guobin Bao (Institute of Pharmacology and Toxicology, UMG) for assisting with protocol establishment and imaging, respectively. We thank Fabian Ludewig (NGS Integrative Genomics Core Unit (NIG), UMG); Christina Weber (Institute of Pharmacology and Toxicology, UMG), Mona Honemann-Capito (Institute of Biochemistry, UMG) for superb technical support. We acknowledge the support by the Open Access Publication Funds of the University of Göttingen. This work was supported by the DFG grant SFB1002 C07 to L.C.Z.; and INF project to Sara Nußbeck and L.C.Z.; the DZHK (German Center for Cardiovascular Research), and the Marie Skłodowska-Curie Actions CRYSTAL3 (Grant agreement ID: 101007931).

## Author contributions
E.S., F.B., G.G. and L.C.Z. are responsible for overall design of this study. E.S., F.B. G.G., C.R., P.T., I.S. and L.C.Z. carried out experiments, data analysis, and interpretation. C.L. analyzed proteomic data. W.M. performed EM. M.Sitte and G.S. preformed SC sequencing and primary QC. Z.V., Z.G. and J.C.G. contribute to EV isolation protocol establishment. F.B. performed bioinformatic analysis. L.C.Z. wrote the paper. E.S., F.B. and G.G. contributed to the preparation and revision of the paper. M.Samak., R.H. and J.C.G. revised the paper. All authors have given approval to the final version of the paper.

## Funding

## Competing interests
The authors declare no competing interests.
