## [Peer Review File · Communications Biology]

Reviewers' comments:

Reviewer #1 (Remarks to the Author):

In this manuscript, Schoger et al. profiled the single-cell transcriptome analysis of hearts with inducible cardiomyocytes-specific Wnt activation, compensatory and failing hypertrophic modeling. The study combines various transcriptomic, proteomic, and biochemical analyses to identify the effects of hypertrophic modeling on cardiomyocytes' adaptive behaviors. The authors provide evidence that upregulated exosome synthesis is found in the hearts of β -cat Δ ex3 mice. They also suggest that the hypertrophic model induces the activation of transcriptomic profiles regarding exosome biogenesis. Lastly, they recruited the iPSCs method to validate the findings through pharmacological perturbation.

While the study provides an appealing phenomenon of altered exosome secretion in hypertrophic models, many single-cell analysis approaches are conjectures producing tools and need independent experiments to validate the hypothesis. Given the enormous literature on adaptive mechanisms in heart hypertrophy, this study would be improved by providing in-depth experimental observations in addition to transcriptomic and proteomic analyses. Although proteomics data to some extent confirms the observations from the transcriptional profiles, it is necessary to run functional assays in justifying the claims. The following outline several suggestions to improve the study.

1. Please illustrate the rationale behind using three different models, including the inducible mouse model, TAC hypertrophic model, and iPSC-derived cardiomyocytes model.
2. In fig. 1, What is the algorithmic standard/characteristics to divide cardiomyocytes into sub-clusters. Is there a difference in the proportion of CM1 and CM2 between control and diseased cells? If there is a difference in CM1 and CM2 percentages, how to explain the difference?
3. Authors use CHIR99021 to increase the Wnt signaling pathway activity and observed increased activity of exosome secretion as shown in iPSC-CMs. This study could be significantly improved by adding Wnt signaling pathway blocker on top of CHIR99021, which eliminates the possible side effects of CHIR99021 in inducing the exosome secretion. In other words, to make a strong correlation between upregulated exosome biogenesis and activated Wnt signaling pathway, it will be worthwhile to study the effects of blunted Wnt signaling pathway on exosome secretion using iPSC-CMs.
4. In fig. 7E, the localization of CRYAB is not significantly different from DMSO to CHIR99021, the difference centers on the intensity of the CRYAB if the same exposure time is used. It would be helpful if the western blotting data can be shown to demonstrate the cytoplasmic accumulation of CRYAB in the CHIR99021 group vs the control.

Reviewer #2 (Remarks to the Author):

Manuscript by Schoger et al investigates EV secretion of cardiomyocytes with canonical Wnt activation by stabilized beta-catenin. The study shows a collection of transcriptomic and proteomic data from mouse and human iPSC-CM models that characterizes differential expression of proteosome constitutes such as CRYAB. The study can be improved with minor changes discussed below.

1. In Figure 1A, the legends and the axis labeling could be enlarged for better readability. It is also difficult to discern the cells from the two conditions with the triangle and circle shapes. Can the authors provide a separate UMAP plot that distinguishes the cells from the two conditions using different colors? Also the conditions are currently labeled "disease" and "healthy" but shouldn't this be "wildtype" and "b-cat exon 3 deletion"?
2. In Fig 1C, are the feature plots representing cells from both CM1 and CM2 clusters (e.g., from all cardiomyocytes identified from each group)? If so, how do the gene expressions look within CM1 cluster and within CM2 cluster separately? Are there other genes that show significant up-/down-regulation in the two distinct CM populations between control vs. exon3 deletion?

3. Labeling in Fig. 2A are too small to read.

4. Please correct spelling errors in figure legend for Fig 5.

5. Most of Wnt-associated genes investigated in the study seem to be canonical Wnt-responsive genes. Have the authors also looked to see if there are any changes in expression of endogenous Wnt ligands, receptors, co-receptors, and the proteins that directly participate in the Wnt signaling pathway?

6. In the Introduction, the authors state that stabilization of beta-catenin (and hence canonical Wnt activation) in cardiomyocytes are deleterious (line 62, page 2). However there exist a myriad of studies that also argue otherwise. As the authors have also shown in Fig. 7 of this manuscript, canonical Wnt activation can lead to CM proliferation as well as other injury repair mechanisms. Therefore this notion should be more delicately discussed.

7. In the Discussion, the authors mention the possibility of detecting EVs in the circulating blood. I wonder if this is truly possible, as I am under the impression that the amount of EVs secreted is not enough for them to be detected in bloodstream.

Reviewer #1 (Remarks to the Author):

In this manuscript, Schoger et al. profiled the single-cell transcriptome analysis of hearts with inducible cardiomyocytes-specific Wnt activation, compensatory and failing hypertrophic modeling. The study combines various transcriptomic, proteomic, and biochemical analyses to identify the effects of hypertrophic modeling on cardiomyocytes' adaptive behaviors. The authors provide evidence that upregulated exosome synthesis is found in the hearts of β -cat Δ ex3 mice. They also suggest that the hypertrophic model induces the activation of transcriptomic profiles regarding exosome biogenesis. Lastly, they recruited the iPSCs method to validate the findings through pharmacological perturbation. While the study provides an appealing phenomenon of altered exosome secretion in hypertrophic models, many single-cell analysis approaches are conjectures producing tools and need independent experiments to validate the hypothesis. Given the enormous literature on adaptive mechanisms in heart hypertrophy, this study would be improved by providing in-depth experimental observations in addition to transcriptomic and proteomic analyses. Although proteomics data to some extent confirms the observations from the transcriptional profiles, it is necessary to run functional assays in justifying the claims. The following outline several suggestions to improve the study.

We thank the reviewer for emphasising this point, which encouraged us to perform further validation in the revised version of our manuscript. Please find below a point-by-point discussion addressing all concerns.

1. Please illustrate the rationale behind using three different models, including the inducible mouse model, TAC hypertrophic model, and iPSC-derived cardiomyocytes model.

The reason of using different experimental models is based on the fact that our study is centred on a hypothesis-driven question based on Wnt signaling activation in cardiac remodeling. Employing a transgenic mouse model with Wnt/B-catenin signaling activation, allows us to precisely study the mechanism driven by Wnt activation in cardiomyocytes and contributing to the cardiac remodeling. Due to the artificial nature of transgenic models, we next compared our finding with a disease experimental TAC hypertrophic model. With these two models, we could should that an EV-mediated response is part of the disease condition. Next, we aimed at confirming that the identified biological process was relevant in cardiomyocytes of human nature. We are aware of the limitations of IPSC-derived cardiomyocytes, and therefore used the system to specifically answer the question whether human derived cardiomyocytes were able to modulate EV-response upon stress caused by Wnt activation. We thank the reviewer for making this point and integrated these arguments in the discussion of the revised manuscript for a better comprehension of the study (lines 391-394).

2. In fig. 1, What is the algorithmic standard/characteristics to divide cardiomyocytes into sub-clusters.

Our entire pipeline is based on the standard Seurat workflow (Stuart and Butler et al. Comprehensive Integration of Single-Cell Data. Cell (2019) [Seurat V3]). This workflow consist of several steps, starting with the normalization of the countmatrix in our assay. We used the NormalizeData tool selecting the standard LogNormalize method. This method uses the gene counts for each cell; divides them by the total cell counts and multiplied by the scale factor. This is then natural-log transformed using log1p. After normalization of the data, we run FindVariableFeatures tool, which identifies the genes that are outliers on a "mean variability plot" using on the "vst" algorithm. This algorithm first fits a line to the relationship of log(variance) and log(mean) using local polynomial regression (loess). Then, it standardizes

the features values (gene expression) using the observed mean and expected variance (given by the fitted line). Feature variance is then calculated on the standardized values after clipping to a maximum. We then scaled and centred the genes in the dataset and regressed out the confounding variables using the ScaleData tool. After that, we reduced the dimensions using RunPCA tool. Here, it first constructs a KNN graph based on the Euclidean distance in PCA space, and then it refines the edge weights between any two cells based on the shared overlap in their local neighbourhoods (Jaccard similarity). To cluster the cells, the tool next applies modularity optimization techniques such as the Louvain algorithm to iteratively group cells together, with the goal of optimizing the standard modularity function. The FindClusters function implements this procedure, and contains a resolution parameter that sets the 'granularity' of the downstream clustering, with increased values leading to a greater number of clusters. We finally visualize the clusters using UMAP. Using this workflow, when analysing all cells together, we were able to visualize one clearly defined CM cluster (based on CM markers). After that, we sub-setted this CM cluster and re-analysed separately. This way, we were able to increase the resolution, observe the different CM subclusters in our CM, and compare them between conditions. This is now adapted in the current revised version for data presented in Fig. 1 and 5.

Is there a difference in the proportion of CM1 and CM2 between control and diseased cells?
If there is a difference in CM1 and CM2 percentages, how to explain the difference?

Yes, there are differences in the CM proportions of the different conditions. In order to illustrate the differences, first differences between all cell types are depicted and secondly, the differences between CM subclusters are subsequently showed in Fig. 1A.

The difference between the proportion of subCM1 and subCM2 may be due to the stress caused by the Wnt signaling activation, which in this model only reached in a subset of CMs (Cre mediated recombination resulting in around 80% of edited cells). When comparing both clusters conditions, we observed that enriched transcripts in the CM1 population categorized to processes related to cell energy and normal cardiac functions, while the CM2 population presented a signature of developmental processes. Thus, CM2 represents a more adaptive cluster, which is increased upon stress condition. This is now integrated in Extended Fig. 2F.

3. Authors use CHIR99021 to increase the Wnt signaling pathway activity and observed increased activity of exosome secretion as shown in iPSC-CMs. This study could be significantly improved by adding Wnt signaling pathway blocker on top of CHIR99021, which eliminates the possible side effects of CHIR99021 in inducing the exosome secretion. In other words, to make a strong correlation between upregulated exosome biogenesis and activated Wnt signaling pathway, it will be worthwhile to study the effects of blunted Wnt signaling pathway on exosome secretion using iPSC-CMs.

This is an excellent observation, which allowed us to expand our conclusions. This experiment has been performed by blocking CHIR-mediated Wnt signaling activation with iso-queracetin (Iso-QC), which specifically blocked the binding of b-catenin and the transcription factor (TCF/LEF) mediating target gene activation¹. In this experiment, we confirmed an increase of the EV marker TSG101 upon CHIR treatment (Wnt activation), whereas a reduction of this marker was observed upon Iso-QC rescue treatment. Additional stress stimulus triggered by TGF- β^2 , showed increased EV marker TSG101 although to a lesser extent. This suggests that stress is the primary cause of increased loading of released EV. This is now included in Fig. 7D and 7E, result lines 259-265 and discussion lines 381-387.

4. In fig. 7E, the localization of CRYAB is not significantly different from DMSO to CHIR99021, the difference centers on the intensity of the CRYAB if the same exposure time is used. It would be helpful if the western blotting data can be shown to demonstrate the cytoplasmic accumulation of CRYAB in the CHIR99021 group vs the control.

This is correct, although we used the same exposure time we cannot depicted difference in quantity with this visualization. For that, we have performed Western blot analysis, which showed no difference of overall protein quantity among the different conditions in whole cell lysate. These data indicated that only the localization of CRYAB is affected upon acute stress. This has been introduced in Fig. 7E and Lines 269-270.

Major changes in the manuscript are highlighted in red and include:

- *Data presentation in Fig. 1, 4, 5, 6 and 7 and Extended Data 2E-G and 10 as well as all the GO analysis*
- *Addition of data in Main Figures: 1G, 4A, 5C, 7C, 7D and 7E and Extended Data: 2G and 2F; 3A, 9B and 10*
- *Cell number was increased for the analysis presented in Fig. 5.*
- *Lines 49-55 and lines 276-292 discuss the deleterious and regenerative roles of Wnt signaling.*
- *Lines 93-99 describe the cell subclustering in the b-catenin stabilization model.*
- *Lines 109-110 describe common GO of downregulated DGEs in CM1 and CM2 of hearts with b-catenin stabilization.*
- *Lines 116-120; 197-200 and 257-259 describe the data showing increased levels of the Ub-proteins in tissue and EVs isolated from Wnt activated hearts as well as in EVs from iPSC-derived cardiomyocyte upon Wnt activation, respectively.*
- *Lines 259-265 and discussion lines 381-387 present the data on IsoQC rescue experiments in iPSC-cardiomyocytes upon Wnt activation.*
- *Lines 269-270 describe the expression of CRYAB in whole cell lysates from iPSC-derived cardiomyocyte upon Wnt activation and rescue*
- *Lines 391-394 argument the use of the different models in the study.*
- *Lines 401-403 discuss possible detection of EVs present in the circulation.*
- *Labeling in the Figures were improved and spelling mistakes were corrected.*
- *The information on the manuscript has been more focused and graphs in Fig. 2 and Fig 7 have been moved to Extended figures.*

We thank the reviewer for the excellent recommendations, which have helped us improve the presentation of our data and extended the validity of our study.

References:

1. Park CH, Chang JY, Hahm ER, Park S, Kim HK and Yang CH. Quercetin, a potent inhibitor against beta-catenin/Tcf signaling in SW480 colon cancer cells. *Biochem Biophys Res Commun.* 2005;328:227-34.
2. Koitabashi N, Danner T, Zaiman AL, Pinto YM, Rowell J, Mankowski J, Zhang D, Nakamura T, Takimoto E and Kass DA. Pivotal role of cardiomyocyte TGF- β signaling in the murine pathological response to sustained pressure overload. *The Journal of clinical investigation.* 2011;121:2301-2312.

Reviewer #2 (Remarks to the Author):

Manuscript by Schoger et al investigates EV secretion of cardiomyocytes with canonical Wnt activation by stabilized beta-catenin. The study shows a collection of transcriptomic and proteomic data from mouse and human iPSC-CM models that characterizes differential expression of proteasome constituents such as CRYAB. The study can be improved with minor changes discussed below.

1. In Figure 1A, the legends and the axis labeling could be enlarged for better readability. It is also difficult to discern the cells from the two conditions with the triangle and circle shapes. Can the authors provide a separate UMAP plot that distinguishes the cells from the two conditions using different colors? Also the conditions are currently labeled "disease" and "healthy" but shouldn't this be "wildtype" and "b-cat exon 3 deletion"?

Thanks for these comments, the figure's labelling and data presentation have been improved accordingly.

2. In Fig 1C, (A) are the feature plots representing cells from both CM1 and CM2 clusters (e.g., from all cardiomyocytes identified from each group)?
(B) If so, how do the gene expressions look within CM1 cluster and within CM2 cluster separately?
(C) Are there other genes that show significant up-/down-regulation in the two distinct CM populations between control vs. exon3 deletion?

(A) Yes, these plots were depicting the two subsets together. In the revised version, we are depicting them separately. We would like to mention, that we recently upgraded our pipeline, which allows reducing even more the background noise and distinguished more clearly the different clusters. The cell filters are the same as used in previous versions, therefore, it gave comparable results by sub-setting the data. This gave more detailed resolution of the CM clusters (CM1 and CM2), which is now presented in Fig. 1 A and B and modified in the text line 93-97.

(B) Accordingly, we illustrated expression of Wnt targets and stress markers in the sub-clusters CM1 and CM2 using dot plots for both conditions in Fig. 1C and 1E. The stress scores is also depicted for CM1 and CM2, which indicated stronger stressed signature in CM2 from b-catenin activated hearts as compared with CM1, in line with the overall Gene Ontology characterizing this subcluster as a more adaptive developmental cluster. Additionally, a Volcano plot of the different subclusters is presented in Extended Fig. 2G and integrated in the text in 97-98.

(C) Yes, with our analysis CM1 showed a total of 144 and CM2 showed 65 dysregulated genes (up/down) ($p < 0.05$ and LogFC 0.25) by comparing control vs. b-catenin stabilization. From this, 23% are commonly dysregulated. The Volcano plot showing the total DGEs is presented in Extended Fig. 2G and the common GO of downregulated DGEs in CM1 and CM2 of b-catenin stabilization is added in Extended Fig. 3A and lines 109-110.

3. Labeling in Fig. 2A are too small to read.
This was adjusted.

4. Please correct spelling errors in figure legend for Fig 5.
This was corrected.

5. Most of Wnt-associated genes investigated in the study seem to be canonical Wnt-responsive genes. Have the authors also looked to see if there are any changes in expression of endogenous Wnt ligands, receptors, co-receptors, and the proteins that directly participate in the Wnt signaling pathway?

We indeed had a look at different Wnt signaling related genes, which do not show major changes among the clusters (shown below). In our transgenic model, we are primarily affecting Wn activation further downstream and therefore we don't expect major changes of upstream Wnt ligands and receptor. Indeed, in our data we observed no major modifications in the expression of endogenous Wnt ligands, receptors or co-receptors when comparing by conditions in cardiomyocytes, endothelial cells, fibroblasts, pericytes, macrophages (CMs, EC, FB, PC and MC). Only neural like cell (NLC) cluster showed upregulation of Fzd7, Fzd6 and Lrp6 and less expression of Fzd3 in Wnt activated hearts. (Figure 1, below). Increase of the Wnt ligands and receptors WNT3A, WNT5A, FZD2 as well as decrease of the Wnt inhibitor DKK3 were previously described in models of hypertrophic remodeling in different animal models¹. We didn't observed major changes in these genes in the hypertrophic model.

Figure 1: Violin plots showing expression levels for Wnt ligands, receptors (Frizzled, Fzd) and co-receptors (Low-density lipoprotein receptor-related protein, Lrp) in different cell types in control (orange) and β -cat^{Δex3} (green) hearts.

6. In the Introduction, the authors state that stabilization of beta-catenin (and hence canonical Wnt activation) in cardiomyocytes are deleterious (line 62, page 2). However there exist a myriad of studies that also argue otherwise. As the authors have also shown in Fig. 7 of this manuscript, canonical Wnt activation can lead to CM proliferation as well as other injury repair mechanisms. Therefore this notion should be more delicately discussed.

This is an excellent comment; we have now complemented introduction and discussion concerning deleterious and regenerative roles of Wnt signaling in the revised version (lines 49-55 and lines 275-289).

We would like to also add that it is a consensus that low Wnt signaling activity maintains homeostasis in the adult heart. However, during cardiac pathologic remodeling components of Wnt signaling become reactivated in multiple cell types including cardiomyocytes¹⁻⁶. Due to contradictory data found by using different models of injury, it is a debate whether b-catenin

signaling should be stimulated or inhibited after myocardial injury¹. What is clear is that b-catenin levels are upregulated in cardiomyocytes that are challenged with various pharmacological and mechanical hypertrophic stimuli, whereas b-catenin stabilization can induce cardiomyocyte hypertrophy and cell cycle activation⁷⁻⁹. In line with these findings, deletion of *Gsk3b*, which results in accumulation of b-catenin, led to substantially induced cell cycle activity of cardiomyocytes accompanied by cardiac-related death in mice¹⁰. Accordingly, cardiomyocyte-specific deletion of b-catenin shows an attenuated TAC-induced hypertrophic remodeling^{9, 11}. We previously showed that b-catenin stabilization and concomitant Wnt transcriptional activation in cardiomyocytes resulted in increased transcription of developmental and cell cycle regulators with cardiomyocytes polynucleation, suggesting endoreduplication, rather than newly formed myocytes and subsequently followed by heart failure⁹. This is similar to the Hippo signaling, which regulates developmental heart growth and proliferation¹² as well as cardiomyocyte dedifferentiation and dysfunction upon long-term activation of the pathway in postnatal cardiomyocytes. This suggests that persistent reactivation of the cell cycle is not beneficial in adult cardiomyocytes. Similar to the approach in the present study, Buikema et al.¹³ induced massive expansion of hiPSC-CMs by glycogen synthase kinase-3 β (GSK-3 β) inhibition using CHIR99021 and concurrent reduction of cell-cell contacts, indicating that additional mechanisms are responsible for cell division arrest and progression to bona fide cell proliferation. Thus, it seems that the activation of the WNT cascade stimulates dedifferentiation and proliferation of mammalian myocytes; however, the response may be determined by environmental and mechanical cues as well as the developmental plasticity of the cardiomyocytes.

7. In the Discussion, the authors mention the possibility of detecting EVs in the circulating blood. I wonder if this is truly possible, as I am under the impression that the amount of EVs secreted is not enough for them to be detected in bloodstream.

This is of course a critical point, which is under investigation. However, previous studies have demonstrated that cardiomyocyte-derived EVs are present in the circulation and that the increased number of cardiac-derived EVs in the blood reflects cardiac injury. It also further confirmed that cellular stress and cardiomyocyte damaging conditions increased EV release, that can serve as biomarker of cardiac injury^{14, 15}. This was introduced in the revised version lines 401-403.

Major changes in the manuscript are highlighted in red and include:

- Data presentation in Fig. 1, 4, 5, 6 and 7 and Extended Data 2E-G and 10 as well as all the GO analysis
- Addition of data in Main Figures: 1G, 4A, 5C, 7C, 7D and 7E and Extended Data: 2G and 2F; 3A, 9B and 10
- Cell number was increased for the analysis presented in Fig. 5.
- Lines 49-55 and lines 276-292 discuss the deleterious and regenerative roles of Wnt signaling.
- Lines 93-99 describe the cell subclustering in the b-catenin stabilization model.
- Lines 109-110 describe common GO of downregulated DGEs in CM1 and CM2 of hearts with b-catenin stabilization.
- Lines 116-120; 197-200 and 257-259 describe the data showing increased levels of the Ub-proteins in tissue and EVs isolated from Wnt activated hearts as well as in EVs from iPSC-derived cardiomyocyte upon Wnt activation, respectively.

- Lines 259-265 and discussion lines 381-387 present the data on IsoQC rescue experiments in iPSC-cardiomyocytes upon Wnt activation.
- Lines 269-270 describe the expression of CRYAB in whole cell lysates from iPSC-derived cardiomyocyte upon Wnt activation and rescue
- Lines 391-394 argument the use of the different models in the study.
- Lines 401-403 discuss possible detection of EVs present in the circulation.
- Labeling in the Figures were improved and spelling mistakes were corrected.
- The information on the manuscript has been more focused and graphs in Fig. 2 and Fig 7 have been moved to Extended figures.

We thank the reviewer for the excellent recommendations, which have helped us improve the presentation of our data and extended the validity of our study.

References:

1. Foulquier S, Daskalopoulos EP, Lluri G, Hermans KCM, Deb A and Blankesteyn WM. WNT Signaling in Cardiac and Vascular Disease. *Pharmacol Rev.* 2018;70:68-141.
2. Kuwahara K and Nakao K. New molecular mechanisms for cardiovascular disease:transcriptional pathways and novel therapeutic targets in heart failure. *J Pharmacol Sci.* 2011;116:337-42.
3. Oerlemans MI, Goumans MJ, van Middelaar B, Clevers H, Doevendans PA and Sluijter JP. Active Wnt signaling in response to cardiac injury. *Basic Res Cardiol.* 2010;105:631-41.
4. Dawson K, Aflaki M and Nattel S. Role of the Wnt-Frizzled system in cardiac pathophysiology: a rapidly developing, poorly understood area with enormous potential. *J Physiol.* 2013;591:1409-32.
5. van de Schans VA, Smits JF and Blankesteyn WM. The Wnt/frizzled pathway in cardiovascular development and disease: friend or foe? *Eur J Pharmacol.* 2008;585:338-45.
6. Hou N, Ye B, Li X, Margulies KB, Xu H, Wang X and Li F. Transcription Factor 7-like 2 Mediates Canonical Wnt/beta-Catenin Signaling and c-Myc Upregulation in Heart Failure. *Circulation Heart failure.* 2016;9.
7. Haq S, Michael A, Andreucci M, Bhattacharya K, Dotto P, Walters B, Woodgett J, Kilter H and Force T. Stabilization of beta-catenin by a Wnt-independent mechanism regulates cardiomyocyte growth. *Proc Natl Acad Sci U S A.* 2003;100:4610-5.
8. Chen X, Shevtsov SP, Hsich E, Cui L, Haq S, Aronovitz M, Kerkelä R, Molkenin JD, Liao R, Salomon RN, Patten R and Force T. The beta-catenin/T-cell factor/lymphocyte enhancer factor signaling pathway is required for normal and stress-induced cardiac hypertrophy. *Mol Cell Biol.* 2006;26:4462-73.
9. Iyer LM, Nagarajan S, Woelfer M, Schoger E, Khadjeh S, Zafiriou MP, Kari V, Herting J, Pang ST, Weber T, Rathjens FS, Fischer TH, Toischer K, Hasenfuss G, Noack C, Johnsen SA and Zelarayan LC. A context-specific cardiac beta-catenin and GATA4 interaction influences TCF7L2 occupancy and remodels chromatin driving disease progression in the adult heart. *Nucleic Acids Res.* 2018;46:2850-2867.
10. Kerkela R, Kockeritz L, Macaulay K, Zhou J, Doble BW, Beahm C, Greytak S, Woulfe K, Trivedi CM, Woodgett JR, Epstein JA, Force T and Huggins GS. Deletion of GSK-3beta in mice leads to hypertrophic cardiomyopathy secondary to cardiomyoblast hyperproliferation. *J Clin Invest.* 2008;118:3609-18.
11. Qu J, Zhou J, Yi XP, Dong B, Zheng H, Miller LM, Wang X, Schneider MD and Li F. Cardiac-specific haploinsufficiency of beta-catenin attenuates cardiac hypertrophy but enhances fetal gene expression in response to aortic constriction. *J Mol Cell Cardiol.* 2007;43:319-26.
12. Heallen T, Zhang M, Wang J, Bonilla-Claudio M, Klysik E, Johnson RL and Martin JF. Hippo pathway inhibits Wnt signaling to restrain cardiomyocyte proliferation and heart size. *Science.* 2011;332:458-61.
13. Buikema JW, Lee S, Goodyer WR, Maas RG, Chirikian O, Li G, Miao Y, Paige SL, Lee D, Wu H, Paik DT, Rhee S, Tian L, Galdos FX, Puluca N, Beyersdorf B, Hu J, Beck A, Venkamatran S, Swami S, Wijnker P, Schuldt M, Dorsch LM, van Mil A, Red-Horse K, Wu JY, Geisen C, Hesse M, Serpooshan V, Jovinge S, Fleischmann BK, Doevendans PA, van der Velden J, Garcia KC, Wu JC, Sluijter JPG and Wu SM. Wnt Activation and Reduced Cell-Cell Contact Synergistically Induce Massive Expansion of Functional Human iPSC-Derived Cardiomyocytes. *Cell stem cell.* 2020;27:50-63.e5.
14. Hegyesi H, Pallinger É, Mecsei S, Hornyák B, Kovácszáci C, Brenner GB, Giricz Z, Pálóczi K, Kittel Á, Tóvári J, Turiak L, Khamari D, Ferdinandy P and Buzás EI. Circulating cardiomyocyte-derived extracellular vesicles reflect cardiac injury during systemic inflammatory response syndrome in mice. *Cell Mol Life Sci.* 2022;79:84.

15. Yarana C, Carroll D, Chen J, Chaiswing L, Zhao Y, Noel T, Alstott M, Bae Y, Dressler EV, Moscow JA, Butterfield DA, Zhu H and St Clair DK. Extracellular Vesicles Released by Cardiomyocytes in a Doxorubicin-Induced Cardiac Injury Mouse Model Contain Protein Biomarkers of Early Cardiac Injury. *Clin Cancer Res.* 2018;24:1644-1653.